# *fmo-4* promotes longevity and stress resistance via ER to mitochondria calcium regulation in *C. elegans*

Angela M Tuckowski[1], Safa Beydoun[2], Elizabeth S Kitto[2], Ajay Bhat[2], Marshall B Howington[1], Aditya Sridhar[3], Mira Bhandari[2], Kelly Chambers[2], Scott F Leiser[2,4]*

[1]Cellular and Molecular Biology Program, University of Michigan, Ann Arbor, United States; [2]Department of Molecular and Integrative Physiology, University of Michigan, Ann Arbor, United States; [3]Department of Molecular, Cellular, and Developmental Biology, University of Michigan, Ann Arbor, United States; [4]Department of Internal Medicine, University of Michigan, Ann Arbor, United States

## eLife Assessment

This **important** study offers **convincing** evidence that *fmo-4* plays essential roles in established lifespan interventions and downstream of its paralog *fmo-2*. The work is of substantial benefit for our understanding of this enzyme family, underscoring their importance in longevity and stress resistance. The study also suggests a connection between *fmo-4* and dysregulation of calcium signalling, with conclusions and interpretations based on **solid** genetic methodology and evidence.

*For correspondence: leiser@umich.edu

**Abstract** Flavin-containing monooxygenases (FMOs) are a conserved family of xenobiotic enzymes upregulated in multiple longevity interventions, including nematode and mouse models. Previous work supports that *C. elegans fmo-2* promotes longevity, stress resistance, and healthspan by rewiring endogenous metabolism. However, there are five *C. elegans* FMOs and five mammalian FMOs, and it is not known whether promoting longevity and health benefits is a conserved role of this gene family. Here, we report that expression of *C. elegans fmo-4* promotes lifespan extension and paraquat stress resistance downstream of both dietary restriction and inhibition of mTOR. We find that overexpression of *fmo-4* in just the hypodermis is sufficient for these benefits, and that this expression significantly modifies the transcriptome. By analyzing changes in gene expression, we find that genes related to calcium signaling are significantly altered downstream of *fmo-4* expression. Highlighting the importance of calcium homeostasis in this pathway, *fmo-4* overexpressing animals are sensitive to thapsigargin, an ER stressor that inhibits calcium flux from the cytosol to the ER lumen. This calcium/*fmo-4* interaction is solidified by data showing that modulating intracellular calcium with either small molecules or genetics can change expression of *fmo-4* and/or interact with *fmo-4* to affect lifespan and stress resistance. Further analysis supports a pathway where *fmo-4* modulates calcium homeostasis downstream of activating transcription factor-6 (*atf-6*), whose knockdown induces and requires *fmo-4* expression. Together, our data identify *fmo-4* as a longevity-promoting gene whose actions interact with known longevity pathways and calcium homeostasis.

## Introduction

Aging is a major risk factor for the onset of cancer, cardiovascular disease, dementia, and many other serious illnesses. Studying the aging process is crucial because a more comprehensive understanding

can lead to the development of therapeutics that treat multiple diseases simultaneously. The use of model organisms, from yeast to mammals, has helped define genetic and environmental pathways that influence aging (*Kenyon, 2010*). Many of these pathways, such as dietary restriction (DR), defined as a decrease in nutrient intake without malnutrition, involve active modification of metabolism that robustly and reproducibly improve health and longevity across species (*Bodkin et al., 2003*; *Swindell, 2012*; *Kapahi et al., 2017*). Similarly, the response to oxygen deficiency, or hypoxic response, can lead to increased longevity, healthspan, and stress resistance (*Shen et al., 2005*). Interestingly, in *C. elegans,* both DR and the hypoxic response converge upon a single gene, *fmo-2,* that is necessary and sufficient to improve health and longevity (*Leiser et al., 2015*).

FMOs are a family of enzymes that use oxygen and NADPH to oxygenate nucleophilic substrates. FMOs oxygenate a wide array of xenobiotic substrates and were discovered ~50 y ago for their role in drug metabolism (*Krueger and Williams, 2005*; *Rossner et al., 2017*). Consequently, a majority of published data on FMOs relate to this role. Recently, studies have begun to focus on the endogenous role(s) of FMOs. Results show that multiple mammalian FMO proteins are involved in systemic metabolism (*Steinbaugh et al., 2012*; *Choi et al., 2023*). We initially discovered a role for *C. elegans fmo-2* in regulating stress resistance and longevity downstream of DR and hypoxia and have continued to identify role(s) for FMO-2 in *C. elegans* metabolism and longevity (*Choi et al., 2023*). However, the conserved mechanisms for FMO-mediated health benefits are still unclear, as are the roles of individual FMO enzymes. Thus, to best understand how FMOs could be leveraged to improve health, we need to understand the conserved mechanism of these enzymes in longevity and metabolism.

There are five *C. elegans* FMOs, and three of the five share significant structural overlap with mammalian FMOs. This overlap involves a predicted endoplasmic reticulum (ER) localization and a membrane-spanning domain, which are each found in *fmo-1, fmo-2*, and *fmo-4*. In comparison to the more well-studied longevity gene *fmo-2, fmo-4* in particular shares 88% identity in the catalytic domain amino acids, suggesting that FMO-2 and FMO-4 may bind similar substrates and could plausibly have overlapping endogenous roles. Published data also show that DR induces both *fmo-2* and *fmo-4* gene expression, implying a role for *fmo-4* in longevity regulation (*Rossner et al., 2017*). Based on this, we hypothesized that *fmo-4* may play a role in *C. elegans* longevity and that studying this role can help to understand this gene family and its relevance to aging.

FMOs are transmembrane enzymes thought to primarily reside in the ER. While mainly studied for its involvement in protein and lipid synthesis, the ER also plays a key role in metabolism and maintaining homeostasis within the cell. For instance, the ER responds to misfolded proteins by eliciting its unfolded protein response (UPR$^{ER}$), thereby removing dysfunctional proteins and restoring homeostasis (*Read and Schröder, 2021*). The ER is also responsible for storing and releasing calcium in a coordinated manner with the cytosol and mitochondria (*Groenendyk et al., 2021*). The ER establishes calcium homeostasis through processes involving (1) sequestering calcium in the lumen with the calreticulin chaperone (*Groenendyk et al., 2021*; *Park et al., 2001*), (2) releasing calcium into the cytosol through the inositol triphosphate receptor (IP$_3$R) (*Groenendyk et al., 2021*; *Berridge, 1993*), and (3) bringing calcium into the ER lumen through the sarcoplasmic endoplasmic reticulum ATPase (SERCA) pump (*Groenendyk et al., 2021*; *Marchi and Pinton, 2014*). Interestingly, the ER's role in regulating metabolic and calcium homeostasis in the cell has recently been linked to longevity in *C. elegans*. Knocking out <u>a</u>ctivating <u>t</u>ranscription <u>f</u>actor (*atf*)–6, one of the three branches of the UPR$^{ER}$, modulates calcium signaling from the ER to the mitochondria, resulting in increased mitochondrial turnover and lifespan extension in *C. elegans* (*Burkewitz et al., 2020*).

Given that (1) *fmo-4* is induced by a longevity-promoting pathway (DR), (2) *fmo-4* is predicted to be an ER transmembrane protein, and (3) the ER's role in cellular homeostasis is linked to aging, we hypothesized that *fmo-4* regulates longevity by modulating ER-related processes. Here, we test this hypothesis by modifying *fmo-4* expression and interrogating interactions between *fmo-4,* stress resistance, longevity, and the ER. Our resulting data show that *fmo-4* promotes longevity and paraquat stress resistance downstream of mTOR and DR through calcium homeostasis. We find that not only is *fmo-4* a regulator of multiple longevity-promoting pathways, but it is also sufficient to extend lifespan and confer paraquat stress resistance when overexpressed either ubiquitously or in the hypodermis. Transcriptomics data and stress assays reveal that calcium signaling is altered downstream of *fmo-4* expression, and that *fmo-4* interacts with calcium regulation between the ER and mitochondria to promote longevity and paraquat stress resistance. Together, these results establish *fmo-4* as a

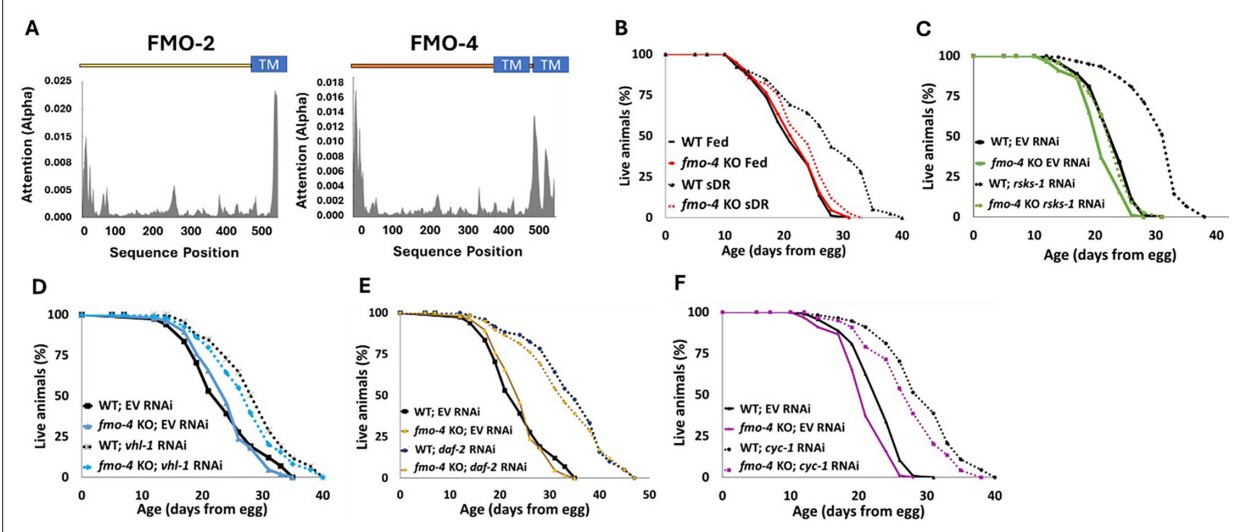

**Figure 1.** *Cefmo-4* regulates mTOR and dietary restriction-mediated longevity. (**A**) Hydropathy plots of FMO-2 and FMO-4 predicting endoplasmic reticulum (ER) transmembrane (TM) domains based on protein sequence analysis. Analysis was done using Deeploc-2.0. (**B**) Lifespan analysis of wild-type (WT) worms and *fmo-4* knockout (*fmo-4* KO) worms on fed and dietary restricted (DR) conditions. (**C**) Lifespan analysis of WT and *fmo-4* KO worms on empty vector (EV) RNAi and *rsks-1* RNAi. (**D**) Lifespan analysis of WT and *fmo-4* KO worms on EV and *vhl-1* RNAi. (**E**) Lifespan analysis of WT and *fmo-4* KO worms on EV and *daf-2* RNAi. (**F**) Lifespan analysis of WT and *fmo-4* KO worms on EV and *cyc-1* RNAi. For each lifespan, n = ~120 worms per condition per experiment, and three replicate experiments were performed. Significance was determined at p<0.05 using log-rank analysis and significant interactions between the condition of interest and genotype were determined at p<0.01 using Cox regression analysis. All replicate data can be found in *Figure 1—source data 1*; *Figure 1—source data 2*; *Figure 1—source data 3*; *Figure 1—source data 4*; *Figure 1—source data 5*.

The online version of this article includes the following source data and figure supplement(s) for figure 1:

**Source data 1.** DR lifespan replicates.

**Source data 2.** mTOR (*rsks-1* RNAi) lifespan replicates.

**Source data 3.** *vhl-1* RNAi lifespan replicates.

**Source data 4.** *daf-2* RNAi lifespan replicates.

**Source data 5.** *cyc-1* RNAi lifespan replicates.

**Figure supplement 1.** Full-length alignment of AncFMO5, Human FMO5, and *C. elegans* FMO-1, FMO-2, and FMO-4.

**Figure supplement 2.** Subcellular localization prediction based on protein sequence of *C. elegans* FMO-4.

longevity-promoting gene and provide evidence as to how *fmo-4* extends *C. elegans* lifespan through calcium-mediated ER to mitochondrial processes.

## Results

### *Cefmo-4* is required for DR and mTOR-mediated lifespan extension

*C. elegans fmo-4* shares significant similarities to its family member and longevity gene, *fmo-2*, including 88% conservation in catalytic residues (*Figure 1—figure supplement 1*), a transmembrane domain (*Figure 1A*) and predicted subcellular localization in the ER (*Figure 1—figure supplement 2*; *Kishore et al., 2020*). Additionally, *fmo-4* is induced by the longevity intervention DR (*Rossner et al., 2017*). These structural similarities and DR-mediated induction led us to hypothesize that *fmo-4* may play a role in aging. To test this, we first asked whether *fmo-4* is required for well-studied longevity pathways, including DR. We utilized *fmo-4(ok294)* knockout (KO) animals (*C. elegans Deletion Mutant Consortium, 2012*) on five conditions reported to extend lifespan in *C. elegans*. Since *fmo-4* is induced by dietary restriction, we started with fed and DR (sDR *Greer and Brunet, 2009*) conditions. Our results show that loss of *fmo-4* has no significant effect on control-fed worms but prevents nearly all the lifespan extension seen under DR (*Figure 1B*). Wild-type (WT) worms on DR experience a ~35% lifespan extension compared to fed WT worms, but when *fmo-4* is knocked out this extension is reduced to ~10% and this interaction is significant by Cox regression (p-value <4.50e$^{-6}$). These data support that *fmo-4* is required for the DR longevity pathway (*Figure 1B*). Having established this role,

we continued lifespan analyses of *fmo-4* KO worms exposed to RNAi knockdown of the S6-kinase gene *rsks-1* (mTOR signaling; *Pan et al., 2007*), the Von Hippel-Lindau gene *vhl-1* (hypoxic signaling; *Leiser et al., 2013*), the insulin receptor *daf-2* (insulin-like signaling; *Kimura et al., 1997*), and the cytochrome c reductase gene *cyc-1* (mitochondrial electron transport chain, cytochrome c reductase; *Lee et al., 2010*; *Figure 1C–F*). The resulting data show that *fmo-4* is fully required for *rsks-1*-mediated longevity as marked by a complete abrogation of the *rsks-1* RNAi lifespan extension when *fmo-4* is knocked out (*Figure 1C*), placing *fmo-4* downstream of mTOR. *fmo-4* was not required for the hypoxic response (*Figure 1D*), insulin-like signaling (*Figure 1E*), or cytochrome c reductase inhibition pathways (*Figure 1F*) to increase lifespan. It is notable that, unlike *fmo-4*, *fmo-2* is required for *vhl-1*-mediated longevity (*Leiser et al., 2015*). This result, coupled with *fmo-4* and *fmo-2* each being required for DR, suggests that these two genes in the same family are overlapping but distinct in their requirement. Together, these data suggest that *fmo-4* plays a necessary role in *C. elegans* longevity regulation downstream of at least two pathways: DR and mTOR signaling.

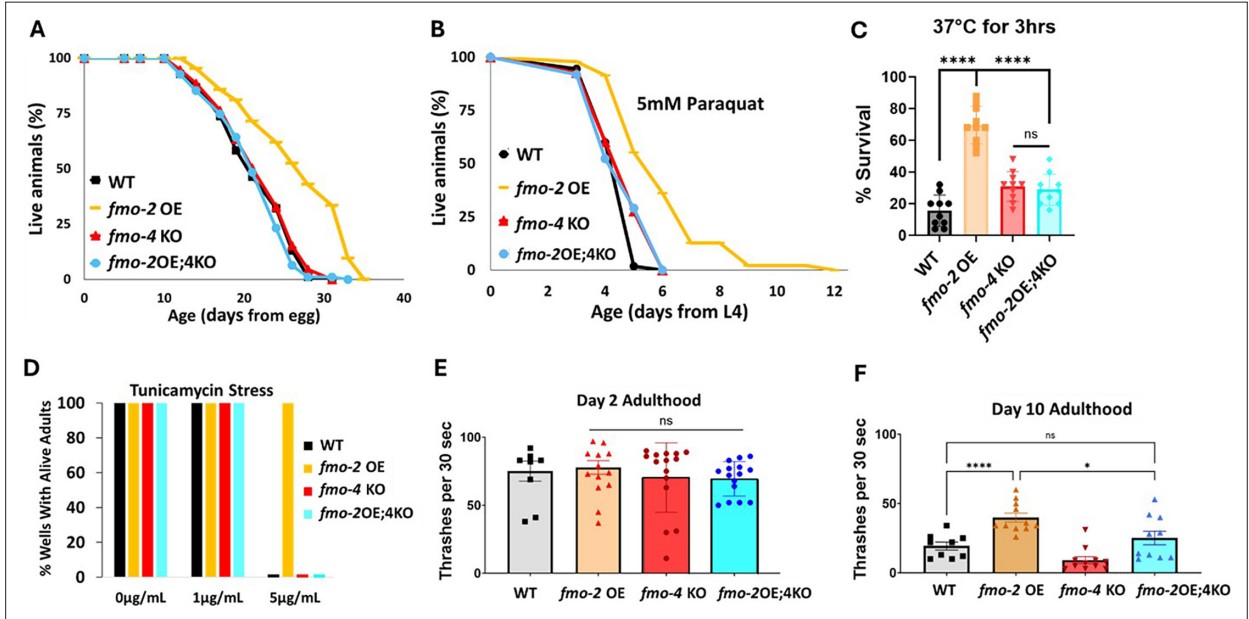

**Figure 2.** *fmo-4* is required for the health benefits of *fmo-2* overexpression. (**A**) Lifespan analysis of wild-type (WT), *fmo-2* overexpressing (*fmo-2* OE), *fmo-4* knockout (*fmo-4* KO), and *fmo-2* OE;*fmo-4* KO (*fmo-2*OE;4KO) worms on *E. coli* OP50 (n = ~120 worms per condition, three replicate experiments). Significance was determined at p<0.05 using log-rank analysis. (**B**) Survival of WT, *fmo-2* OE, *fmo-4* KO, and *fmo-2*OE;4KO worms exposed to 5 mM paraquat at L4 stage (n = ~90 worms per condition, three replicate experiments). Significance was determined at p<0.05 using log-rank analysis. (**C**) Survival of worms exposed to 37 °C heat shock for 3 hr (hours) at L4 stage (n=100 worms per condition, three replicate experiments). (**D**) Survival of worms exposed to 0, 1, and 5 µg/mL tunicamycin from the egg until day 1 of adulthood (n = ~60 eggs per condition, three replicate experiments). (**E**) Healthspan analysis of worm thrashing in a drop of M9 solution for 30 s on day 2 of adulthood (n=10 worms per condition, three replicate experiments). (**F**) Healthspan analysis of worm thrashing in a drop of M9 solution for 30 s on day 10 of adulthood (n=10 worms per condition, three replicate experiments). For heat stress, tunicamycin stress, and healthspan assessments, * denotes significant change at p<0.05 using unpaired two-tailed t-test or one-way ANOVA. N.S. = not significant. All replicate data can be found in *Figure 2—source data 1*; *Figure 2—source data 2*; *Figure 2—source data 3*; *Figure 2—source data 4*; *Figure 2—source data 5*.

The online version of this article includes the following source data for figure 2:

**Source data 1.** *fmo-2* OE;4KO lifespan replicates.

**Source data 2.** *fmo-2* OE;4KO paraquat stress replicates.

**Source data 3.** *fmo-2* OE;4KO heat stress replicates.

**Source data 4.** *fmo-2* OE;4KO tunicamycin stress replicates.

**Source data 5.** *fmo-2* OE;4KO healthspan replicates.

## *fmo-4* is required for *fmo-2*-mediated longevity, stress resistance, and healthspan

Having established that, like *fmo-2*, *fmo-4* is required for DR-mediated longevity, we next hypothesized that *fmo-2* and *fmo-4* may be acting in the same pathway. To test this, we crossed *fmo-2* overexpressing (OE) worms with *fmo-4* KO worms to create an *fmo-2* OE; *fmo-4* KO strain (*fmo-2*OE;4KO). We validated this strain via qPCR analysis (*Supplementary file 1*). Upon measuring their lifespan compared to *fmo-2* OE animals, we find that knocking out *fmo-4* completely abrogates the lifespan extension from *fmo-2* OE (*Figure 2A*). This result suggests that *fmo-4* is required for *fmo-2*-mediated longevity and thus may act downstream of *fmo-2* to promote longevity in *C. elegans*.

We previously showed that *fmo-2* OE is not just sufficient for lifespan extension but is also sufficient to promote resistance to multiple forms of stress, including oxidative stress (paraquat), heat stress, and ER stress (tunicamycin; *Leiser et al., 2015*). Since *fmo-4* is required for *fmo-2*-mediated longevity, we hypothesized that *fmo-4* would also be required for *fmo-2*-mediated stress resistance. Following exposure to 5 mM paraquat, 37 °C heat stress, or 5 µg/mL tunicamycin, we find that the *fmo-2*OE;4KO worms survive similarly to WT worms and are no longer resistant to these stresses (*Figure 2B–D*). Thus, knocking out *fmo-4* blocks *fmo-2*-mediated broad stress resistance.

Healthspan is a crucial component to longevity, and we previously found that *fmo-2* OE is sufficient to promote healthspan benefits with age in *C. elegans* (*Leiser et al., 2015*). As *fmo-4* is required for both the lifespan extension and stress resistance observed with *fmo-2* OE, we hypothesized that *fmo-4* would also be required for *fmo-2*-mediated healthspan benefits. To test this, we measured the thrashing rate of *fmo-2* OE;4KO worms at day 2 and day 10 of adulthood. We find that knocking out *fmo-4* abrogates the healthspan benefits of the *fmo-2* OE worms with age (*Figure 2E, F*). Thus, *fmo-4* is required for *fmo-2*-mediated health and resilience benefits, further supporting its acting downstream of *fmo-2* to promote health and longevity.

## Overexpression of *fmo-4* promotes longevity, healthspan, and paraquat stress resistance

While being necessary for increased healthspan, lifespan, and stress resistance shows that *fmo-4* is required for these benefits, it does not test whether it can modulate aging directly. Therefore, to better understand *fmo-4's* role in longevity and health, we next asked whether *fmo-4* is sufficient for lifespan extension, stress resistance, and healthspan benefits. We created a ubiquitous *fmo-4* OE worm strain with *fmo-4* expressed under the *eft-3* promoter via multicopy extrachromosomal array followed by random integration. qPCR analysis shows that *fmo-4* is overexpressed ~150 fold over WT in the ubiquitous *fmo-4* OE worms (*Supplementary file 1*), and their development time is slightly (~1.5 hr) but statistically significantly delayed compared to WT (*Figure 3—figure supplement 1*). Interestingly, we find that ubiquitous *fmo-4* OE is sufficient to reproducibly extend lifespan by ~20% (*Figure 3A*). Epistatic analysis with *fmo-2* shows that *fmo-4* OE lifespan extension is not additive with *fmo-2* OE (*Figure 3—figure supplement 2A*), as determined by comparing the individual OE strains to the *fmo-2* OE;*fmo-4* OE (*fmo-2*OE;4 OE) worm strain (*Supplementary file 1*). This further supports that these two genes in the same family act in the same pathway. To test whether *fmo-4* requires *fmo-2*, we overexpressed *fmo-4* in the context of *fmo-2* knockout (*fmo-4* OE;2KO) (*Supplementary file 1*). The results show that *fmo-4* OE does not require *fmo-2* for its longevity benefits (*Figure 3—figure supplement 3A*). This further validates that *fmo-4* likely acts downstream of *fmo-2*.

Increased longevity is frequently observed in tandem with increased healthspan and resistance to toxic stress (*Soo et al., 2023*). We assessed the healthspan of the ubiquitous *fmo-4* OE worms at days 2 and 10 of adulthood by measuring thrashing rates (movement). We find that ubiquitous *fmo-4* OE is sufficient for an improvement (p=0.011) in healthspan benefits with age (*Figure 3B, C*). We next measured the stress resistance of the ubiquitous *fmo-4* OE worms to 5 mM paraquat (oxidative stress), 37 °C heat stress, 5 µg/mL of tunicamycin (ER glycosylation stress), and 1 mg/mL thapsigargin (ER calcium stress). We included two ER stresses because FMO-4 is predicted to be an ER transmembrane protein. We find that ubiquitous *fmo-4* OE is sufficient to convey paraquat stress resistance (*Figure 3D*) but does not offer any protection from heat (*Figure 3E*) or tunicamycin stress (*Figure 3F*). Interestingly, ubiquitous *fmo-4* overexpressing animals are sensitive to thapsigargin, as shown by the worms' significantly diminished size and inability to develop compared to WT control worms (*Figure 3G–J*). These results suggest a narrow stress resistance and even some sensitivity for

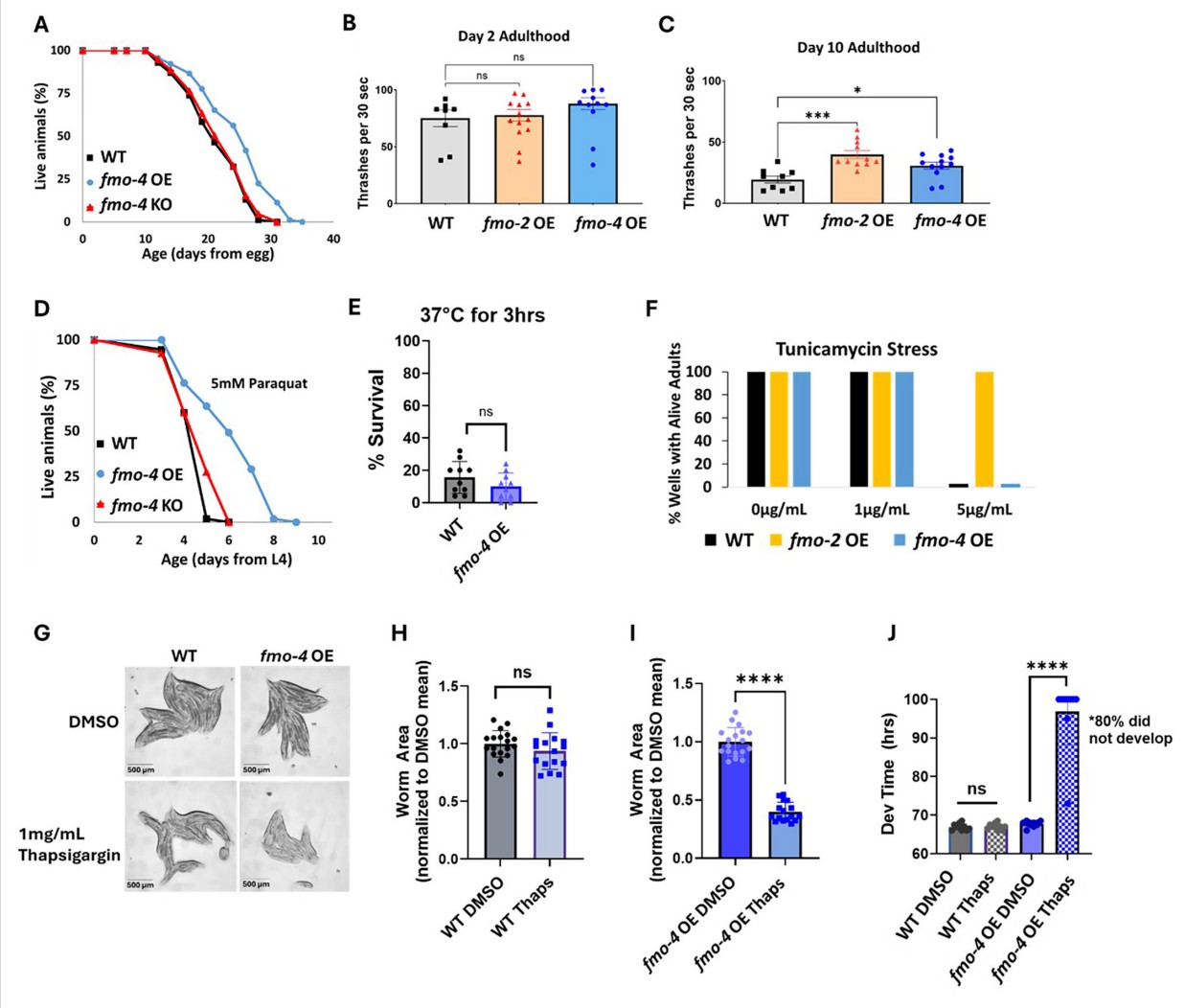

**Figure 3.** Overexpressing *fmo-4* is sufficient for lifespan extension, paraquat stress resistance, and improved healthspan. (**A**) Lifespan analysis of wild-type (WT), *fmo-4* overexpressing (*fmo-4* OE), and *fmo-4* knockout (*fmo-4* KO) worms on *E. coli* OP50 (n = ~120 worms per condition, three replicate experiments). Significance was determined at p<0.05 using log-rank analysis. (**B**) Healthspan analysis of worms thrashing in a drop of M9 solution for 30 s (seconds) on day 2 of adulthood (n=10 worms per condition, three replicate experiments). (**C**) Healthspan analysis of worms thrashing in a drop of M9 solution for 30 s on day 10 of adulthood (n=10 worms per condition, three replicate experiments). (**D**) Survival of worms exposed to 5 mM paraquat starting from L4 stage (n = ~90 worms per condition, three replicate experiments). Significance was determined at p<0.05 using log-rank analysis. (**E**) Survival of worms exposed to 37 °C for 3 hr (hours) at L4 stage (n=100 worms per condition, three replicate experiments). (**F**) Survival of worms exposed to 0, 1, and 5 µg/mL tunicamycin starting from the egg until day 1 of adulthood (n = ~60 eggs per condition, three replicate experiments). (**G**) Brightfield images of WT and ubiquitous *fmo-4* OE worms exposed to DMSO or 1 mg/mL thapsigargin (n = ~20 worms per condition, three replicate experiments). Quantifications of (**G**) images in (**H, I**). (**J**) Development (Dev) time in hours (hrs) of WT and *fmo-4* OE worms grown on DMSO control or 1 mg/mL thapsigargin (Thaps) from L2 stage (n=~10 worms per condition, three replicate experiments). * denotes significant change at p<0.05 using t-test. N.S. = not significant. All replicate data can be found in *Figure 3—source data 1*; *Figure 3—source data 2*; *Figure 3—source data 3*; *Figure 3—source data 4*; *Figure 3—source data 5*; *Figure 3—source data 6*; *Figure 3—source data 7*.

The online version of this article includes the following source data and figure supplement(s) for figure 3:

**Source data 1.** *mo-4* OE ubuitous lifespan replicates.

**Source data 2.** *fmo-4* OE ubiquitous healthspan replicates.

**Source data 3.** *fmo-4* OE ubuitous paraquat stress replicates.

**Source data 4.** *fmo-4* OEubuitous heat stress replicates.

**Source data 5.** *fmo-4* OE ubuitous tunicamycin stress replicates.

**Source data 6.** *fmo-4* OE ubuitous thapsigargin stress replicates.

*Figure 3 continued on next page*

*Figure 3 continued*

**Source data 7.** *fmo-4* OE ubuitous thapsigargin development replicates.

**Figure supplement 1.** Development time of experimental worm strains compared to wild-type.

**Figure supplement 1—source data 1.** Replicates of development time of experimental worm strains compared to wild-type.

**Figure supplement 2.** *fmo-4* acts in the same pathway as *fmo-2*.

**Figure supplement 2—source data 1.** *fmo-2* OE;4OE lifespan replicates.

**Figure supplement 2—source data 2.** *fmo-2* OE;4OE paraquat stress replicates.

**Figure supplement 2—source data 3.** *fmo-2* OE;4OE heat stress replicates.

**Figure supplement 2—source data 4.** *fmo-2* OE;4OE tunicamycin stress replicates.

**Figure supplement 2—source data 5.** *fmo-2* OE;4OE thapsigargin stress replicates.

**Figure supplement 3.** Knocking out *fmo-2* does not affect *fmo-4* overexpression.

**Figure supplement 3—source data 1.** *fmo-4* OE;2KO lifespan replicates.

**Figure supplement 3—source data 2.** *fmo-4* OE;2KO paraquat stress replicates.

**Figure supplement 3—source data 3.** *fmo-4* OE;2KO heat stress replicates.

**Figure supplement 3—source data 4.** *fmo-4* OE;2KO tunicamycin stress replicates.

**Figure supplement 3—source data 5.** *fmo-4* OE;2KO thapsigargin stress replicates.

*fmo-4* in comparison to other longevity genes. To test whether *fmo-4* OE would interact with *fmo-2* OE, we combined the strains and measured their stress resistance. We find that the *fmo-2* OE strain is still resistant to paraquat, heat, and tunicamycin stress in the presence of increased *fmo-4*, suggesting that overexpression of *fmo-4* does not affect resistance to these stresses (*Figure 3—figure supplement 2B–D*). Interestingly, the *fmo-2* OE strain is sensitive to thapsigargin stress in the presence of increased *fmo-4*, suggesting that *fmo-4* likely also acts downstream of *fmo-2* in this stress pathway (*Figure 3—figure supplement 2E–I*). Overall, these results suggest that *fmo-4* (1) is sufficient to improve healthspan, (2) is both necessary and sufficient to promote paraquat resistance, (3) is necessary but not sufficient to improve heat and tunicamycin resistance, and (4) promotes sensitivity to calcium-mediated ER stress from thapsigargin. As expected under the hypothesis that *fmo-4* acts downstream of *fmo-2,* the *fmo-4*OE;2KO strain phenocopies the *fmo-4* OE strain in all stress assays (*Figure 3—figure supplement 3B–I*). Together, these data suggest that *fmo-4* OE is sufficient to promote lifespan extension and paraquat stress resistance as well as a healthspan benefit, while exhibiting sensitivity to ER calcium stress.

## Hypodermal overexpression of *fmo-4* is sufficient for longevity and paraquat resistance

While using the ubiquitous overexpressing worm strain is helpful to probe into *fmo-4's* involvement in longevity, stress resistance, and healthspan, it is also important to note that *fmo-4* is normally localized and expressed primarily in the hypodermis (*Figure 4A–C*; *Petalcorin et al., 2005*). Thus, we asked whether hypodermal-specific *fmo-4* overexpression is sufficient for these health benefits. We created a hypodermal-specific *fmo-4* OE worm strain expressing *fmo-4* under the *dpy-7* promoter via multicopy extrachromosomal arrays followed by random integration. qPCR data show that *fmo-4* is expressed ~45 fold over WT in the hypodermal-specific *fmo-4* OE worms (*Supplementary file 1*), and their developmental time does not differ from WT (*Figure 3—figure supplement 1*). We measured lifespan, resistance to paraquat, heat, tunicamycin, and thapsigargin stress, as well as the healthspan of the hypodermal-specific *fmo-4* OE worms (*fmo-4* OE^Hyp^). We find that the *fmo-4* OE^Hyp^ strain exhibits a lifespan extension (*Figure 4D*), resistance to paraquat (*Figure 4E*) but not heat or tunicamycin stress (*Figure 4F, G*), and sensitivity to thapsigargin stress (*Figure 4H–J*). However, the *fmo-4* OE^Hyp^ strain did not have a statistically significant effect on healthspan unlike the ubiquitous *fmo-4* OE strain (*Figure 4K, L*). Overall, these results agree with the *C. elegans* single-cell atlas as well as previously published work (*Petalcorin et al., 2005*) showing that *fmo-4* is mostly localized to the hypodermis (*Figure 4A*) and is consistent with the ubiquitous strain primarily benefiting from expressing additional *fmo-4* in the hypodermis. Thus, ubiquitous or hypodermal-specific overexpression of *fmo-4* is sufficient for longevity and paraquat stress resistance (*Figures 3A, D and 4D, E*),

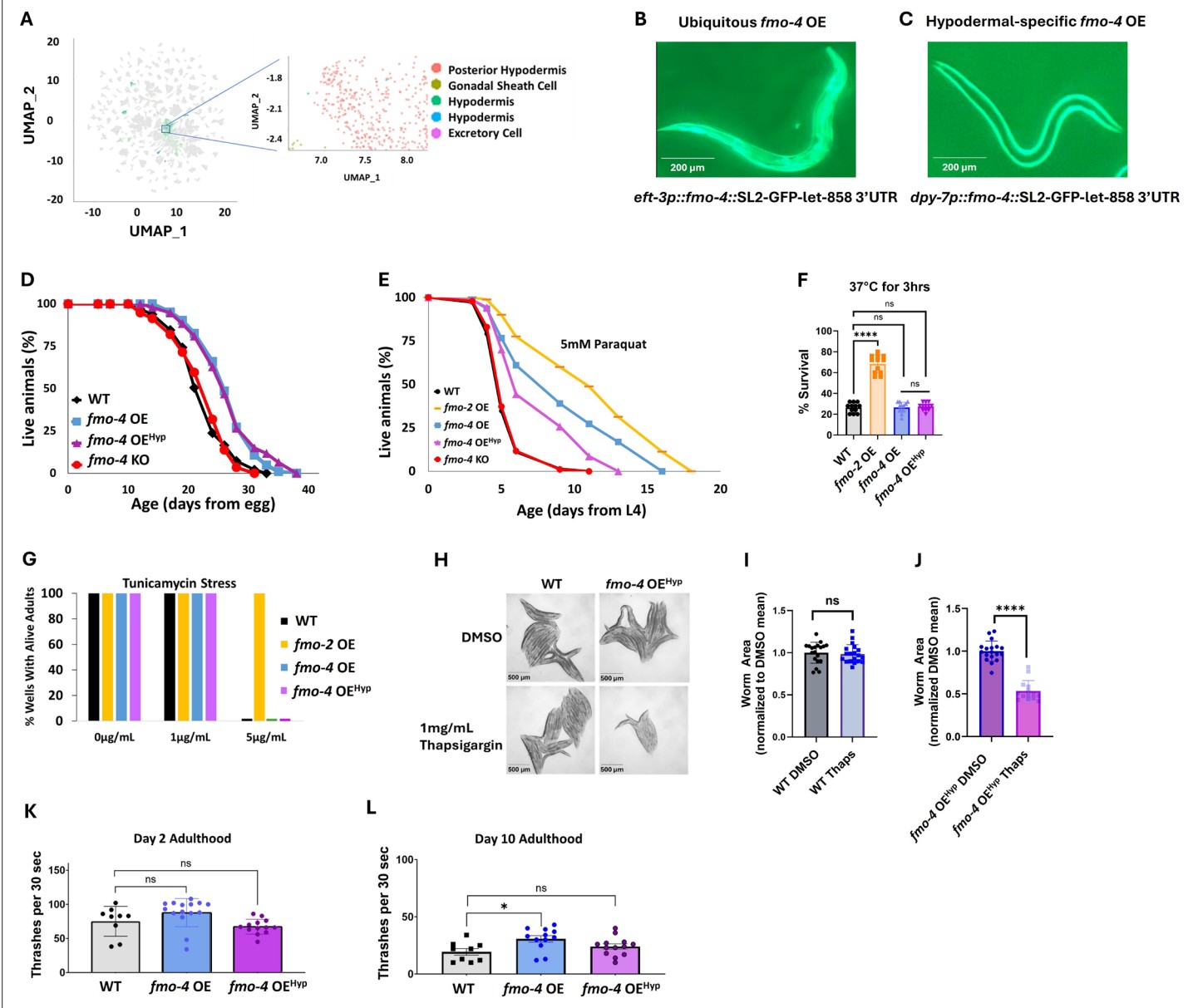

**Figure 4.** Overexpressing *fmo-4* in the hypodermis is sufficient for lifespan extension and paraquat stress resistance. (**A**) Complete cell atlas of *C. elegans* aging cluster map highlighting regions of *fmo-4* gene expression. (**B**) Image of the ubiquitous *fmo-4* overexpressing (*fmo-4* OE) worm by the *eft-3* promoter showing expression throughout its body. (**C**) Image of the hypodermal-specific *fmo-4* OE worm by the *dpy-7* promoter showing expression in the hypodermis. (**D**) Lifespan analysis of wild-type (WT), *fmo-4* OE, hypodermal-specific *fmo-4* OE (*fmo-4* OE^Hyp), and *fmo-4* knockout (*fmo-4* KO) worms (n = ~120 worms per condition, three replicate experiments). Significance was determined at p<0.05 using log-rank analysis. (**E**) Survival of worms exposed to 5 mM paraquat starting at L4 stage (n = ~90 worms per condition, three replicate experiments). Significance was determined at p<0.05 using log-rank analysis. (**F**) Survival of worms exposed to 37 °C heat for 3 hr (hours) at L4 stage (n=100 worms per condition, three replicate experiments). (**G**) Survival of worms exposed to 0, 1, and 5 ug/mL tunicamycin starting at egg until day 1 of adulthood (n = ~60 eggs per condition, three replicate experiments). (**H**) Brightfield images of WT and *fmo-4* OE^Hyp worms exposed to DMSO or 1 mg/mL thapsigargin (n = ~20 worms per condition, three replicate experiments). Quantification of (**H**) in (**I–J**). (**K**) Healthspan analysis of worms thrashing in a drop of M9 solution for 30 s (seconds) on day 2 of adulthood (n=10 worms per condition, three replicate experiments). (**L**) Healthspan analysis of worms thrashing in a drop of M9 solution for 30 s on day 10 of adulthood (n=10 worms per condition, three replicate experiments). For heat stress, tunicamycin stress, thapsigargin stress, and healthspan assessments, * denotes significant change at p<0.05 using t-test. N.S. = not significant. All replicate data can be found in *Figure 4—source data 1*; *Figure 4—source data 2*; *Figure 4—source data 3*; *Figure 4—source data 4*; *Figure 4—source data 5*; *Figure 4—source data 6*.

The online version of this article includes the following source data for figure 4:

**Source data 1.** *fmo-4* OE hypodermal lifespan replicates.

*Figure 4 continued on next page*

*Figure 4 continued*

**Source data 2.** *fmo-4* OE hypodermal paraquat stress replicates.

**Source data 3.** *fmo-4* OE hypodermal heat stress replicates.

**Source data 4.** *fmo-4* OE hypodermal tunicamycin stress replicates.

**Source data 5.** *fmo-4* OE hypodermal thapsigargin stress replicates.

**Source data 6.** *fmo-4* OE hypodermal healthspan replicates.

does not affect heat or tunicamycin resistance (*Figure 3E, F*, *Figure 4F, G*), and results in thapsigargin sensitivity (*Figures 3G–J and 4H–J*).

## *fmo-4* OE transcriptomics reveals a link to calcium regulation

To identify the downstream effects of *fmo-4* expression, we analyzed the transcriptome of *fmo-4* OE animals and compared them with control animals (*Figure 5A*). Based on a p-value of <0.05, we find ~800 transcripts are upregulated and ~500 transcripts are downregulated when *fmo-4* is overexpressed (*Source data 1*). Of the upregulated and downregulated transcripts unique to the *fmo-4* OE profile, we noticed that some of the significant pathways include transcription, Wnt signaling, TGFβ signaling, protein processing in the ER, and other subsets of signaling, as determined by the **D**atabase for **A**nnotation, **V**isualization and **I**ntegrated **D**iscovery (DAVID) Functional Annotation Bioinformatics Microarray Analysis (*Figure 5A*). Based on its known role as a xenobiotic enzyme, we were intrigued to find that *fmo-4* OE is modulating signaling pathways. We were further intrigued to find that one of the signaling pathways affected by *fmo-4* is calcium signaling (*Source data 2*). This was interesting considering that *fmo-4* OE worms are sensitive to thapsigargin (*Figure 3G–J*), implicating an interaction between calcium signaling and *fmo-4*. To further verify this possibility, we used another tool called PANTHER Classification System Protein Class, which identified FMO-4 as a putative transmembrane signal receptor (G-protein coupled receptor), ion transporter, and/or calcium-binding protein (*Source data 2*). Together, these results support the possibility that FMO-4 is involved in calcium-related processes.

To begin exploring this interaction, we determined how changes in intracellular calcium levels impact *fmo-4* expression and lifespan. We first manipulated calcium levels by supplementing the acetylcholine agonist, carbachol. Carbachol activates acetylcholine receptors, increasing overall intracellular calcium levels (*Masoumi et al., 2023*). We first validated that carbachol increases calcium levels by utilizing worms that neuronally express the calcium indicator GCaMP7f (SWF702; *Dag et al., 2023*). When these worms were exposed to 300 µM carbachol, a significant increase in GFP fluorescence was reported, indicating an increase in calcium levels (*Figure 5—figure supplement 1A–D*). We then measured *fmo-4* gene expression upon exposure to carbachol using an *fmo-4p::mCherry* transcriptional reporter strain. Interestingly, we find that supplementation of 300 µM carbachol induces *fmo-4* promoter expression fluorescence nearly twofold over a water control (*Figure 5B, C*). This was similar to the level of induction of *fmo-4* in fasted (DR-like) conditions (*Figure 5B, C*). We postulate that *fmo-4* is induced by carbachol because increased intracellular calcium activates *fmo-4* gene expression. Since carbachol supplementation induces *fmo-4*, we hypothesized that carbachol may affect lifespan and would interact with *fmo-4*. We measured the lifespan of worms supplemented with 50 µM carbachol and find that while WT lifespan is significantly extended by carbachol, this extension is not additive with *fmo-4* OE (*Figure 5D*), and requires *fmo-4,* as the *fmo-4* KO is shorter-lived when exposed to carbachol (*Figure 5E*).

To test whether reducing calcium has an opposing effect to increasing it, we utilized EthyleneDiamineTetraAcetic acid (EDTA) to deplete calcium levels (*Mellau and Jørgensen, 2003*). We validated that EDTA depletes calcium levels by exposing GCaMP7f expressing worms (*Dag et al., 2023*) to 10 mM EDTA and we observed a significant decrease in GFP fluorescence, indicating a decrease in calcium levels (*Figure 5—figure supplement 1A, B, E, F*). Then, we placed *fmo-4p::mCherry* reporter worms on plates containing 10 mM EDTA and measured fluorescence. Surprisingly, we find that EDTA also significantly induces *fmo-4* expression (*Figure 5B, C*). We then assessed the survival of WT, *fmo-4* OE and KO worms on plates supplemented with 50 µM EDTA. We find that EDTA significantly extends WT lifespan without further extending *fmo-4* OE lifespan (*Figure 5F*), while shortening the lifespan of *fmo-4* KO animals (*Figure 5G*). Overall, these data suggest that *fmo-4* is highly sensitive to changing

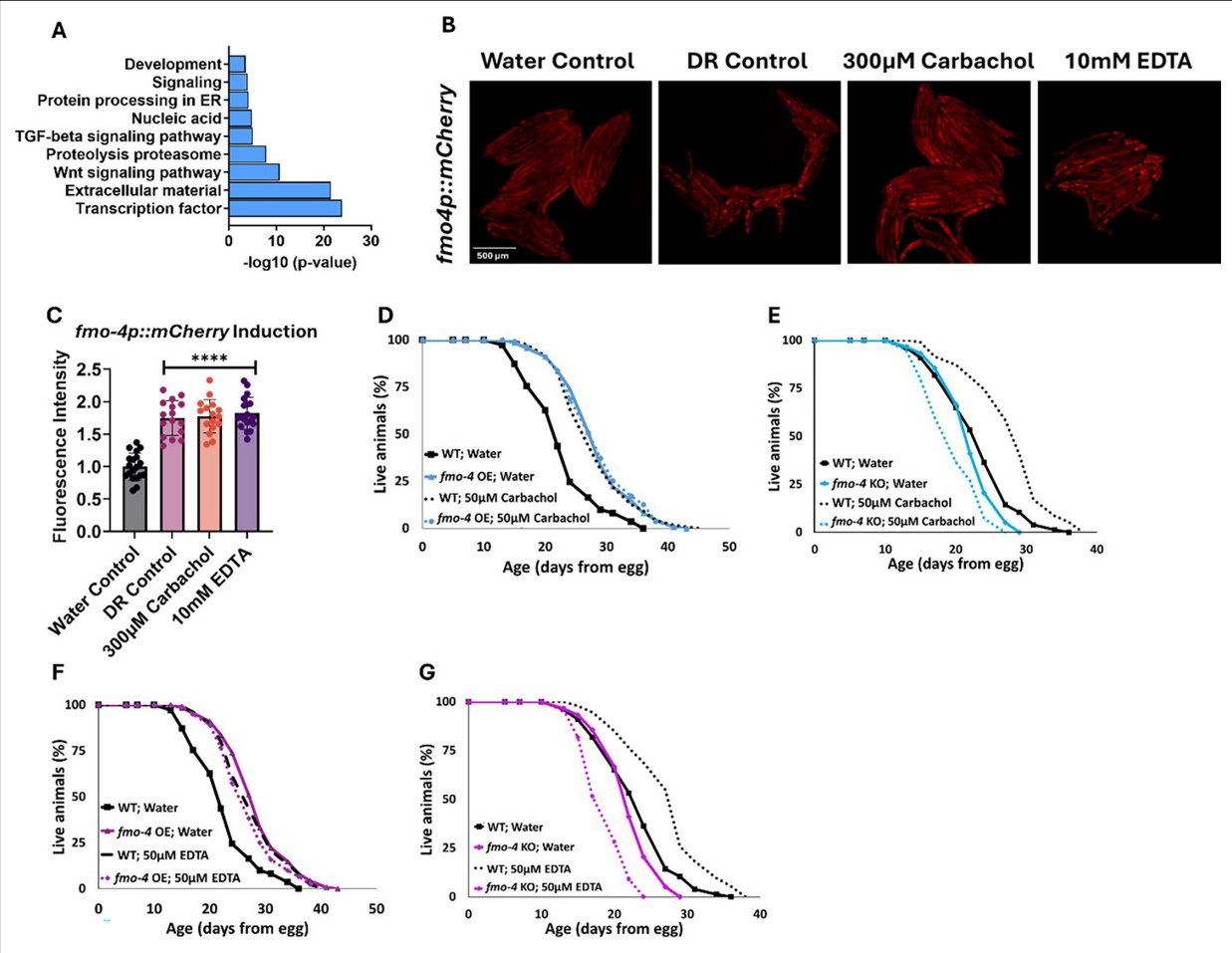

**Figure 5.** *fmo-4* OE transcriptomics reveals a link to calcium regulation. (**A**) Gene ontology (GO) analysis of significantly regulated pathways unique to *fmo-4* overexpression (*fmo-4* OE). (**B**) Fluorescence images of *fmo-4p::mCherry* reporter worms exposed to a water control, dietary restriction (DR) control, 300 μM carbachol, or 10 mM EDTA, which is quantified in (**C**) (n = ~20 worms per condition, three replicate experiments). (**D**) Lifespan assessment of wild-type (WT) and *fmo-4* OE worms exposed to a water control or 50 μM carbachol. (**E**) Lifespan assessment of WT and *fmo-4* knockout (*fmo-4* KO) worms exposed to a water control or 50 μM carbachol (**F**) Lifespan assessment of WT and *fmo-4* OE worms exposed to a water control or 50 μM EDTA. (**G**) Lifespan assessment of WT and *fmo-4* KO worms exposed to a water control or 50 μM EDTA. For all lifespan analyses, n = ~120 worms per condition, three replicate experiments performed. Significance was determined at p<0.05 using log-rank analysis and significant interactions between the condition of interest and genotype was determined at p<0.01 using Cox regression analysis. For imaging experiments, * denotes significant change at p<0.05 using unpaired two-tailed t-test. N.S. = not significant. All replicate data can be found in *Figure 5—source data 1*; *Figure 5—source data 2*; *Figure 5—source data 3*.

The online version of this article includes the following source data and figure supplement(s) for figure 5:

**Source data 1.** Carbachol and EDTA imaging replicates.

**Source data 2.** Quantification of Carbachol and EDTA imaging replicates.

**Source data 3.** Carbachol and EDTA lifespan replicates.

**Figure supplement 1.** Carbachol and EthyleneDiamineTetraAcetic acid (EDTA) alter GCaMP7f fluorescence intesity.

**Figure supplement 1—source data 1.** GCaMP7f on Carbachol imaging replicates.

**Figure supplement 1—source data 2.** GCaMP7f on EDTA imaging replicates.

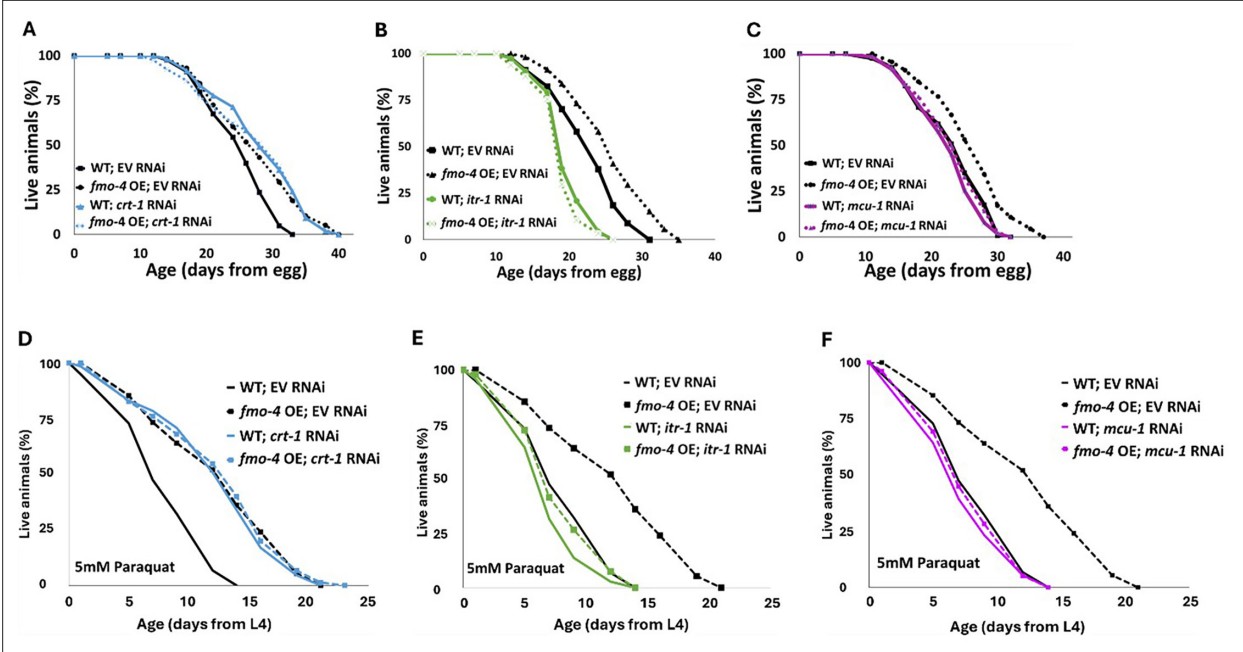

**Figure 6.** *fmo-4* interacts with calcium signaling genes to promote longevity and paraquat resistance. (**A**) Survival analysis of wild-type (WT) and *fmo-4* overexpressing (*fmo-4* OE) worms on empty vector (EV) and *crt-1* RNAi. (**B**) Survival analysis of worms on EV and *itr-1* RNAi. (**C**) Survival analysis of worms on EV and *mcu-1* RNAi. For all lifespan analyses, n = ~120 worms per condition, three replicate experiments were performed. (**D**) Survival of WT and *fmo-4* OE worms on EV and *crt-1* RNAi exposed to 5 mM paraquat at L4 stage. (**E**) Survival of worms on EV and *itr-1* RNAi exposed to 5 mM paraquat at L4 stage. (**F**) Survival of worms on EV and *mcu-1* RNAi exposed to 5 mM paraquat at L4 stage. For all paraquat survival assays, n = ~90 worms per condition, three replicate experiments performed. Significance was determined at p<0.05 using log-rank analysis and significant interactions between the condition of interest and genotype were determined at p<0.01 using Cox regression analysis. All replicate data can be found in *Figure 6—source data 1*; *Figure 6—source data 2*; *Figure 6—source data 3*; *Figure 6—source data 4*.

The online version of this article includes the following source data and figure supplement(s) for figure 6:

**Source data 1.** *fmo-4* OE worms on *crt-1* RNAi lifespan replicates.

**Source data 2.** *fmo-4* OE worms on *itr-1* RNAi lifespan replicates.

**Source data 3.** *fmo-4* OE worms on *mcu-1* RNAi lifespan replicates.

**Source data 4.** Paraquat stress replicates of *fmo-4* OE worms on *crt-1*, *itr-1* and *mcu-1* RNAi.

**Figure supplement 1.** Lifespan and paraquat survival assessments of *fmo-4* KO worms on *crt-1, itr-1,* and *mcu-1* RNAi.

**Figure supplement 1—source data 1.** *fmo-4* KO worms on *crt-1* RNAi lifespan replicates.

**Figure supplement 1—source data 2.** *fmo-4* KO worms on *itr-1* RNAi lifespan replicates.

**Figure supplement 1—source data 3.** *fmo-4* KO worms on *mcu-1* RNAi lifespan replicates.

**Figure supplement 1—source data 4.** Paraqaut stress assays of *fmo-4* KO worms on *crt-1, itr-1, mcu-1,* and *vdac-1* RNAi replicates.

calcium levels in either direction, and that both increasing and depleting calcium (1) induces *fmo-4*, (2) extends WT lifespan, (3) is not additive with *fmo-4* OE lifespan, and (4) is deleterious to *fmo-4* KO animals. Thus, these results support an interaction between *fmo-4* and calcium signaling.

## *fmo-4* genetically interacts with calcium signaling to promote longevity and paraquat resistance

Having established an interaction between *fmo-4* and calcium perturbations, we next asked if *fmo-4* interacts with genes involved in calcium signaling. These genes include calreticulin (*crt-1*), inositol triphosphate receptor (IP₃R, *itr-1*), and the mitochondrial calcium uniporter (*mcu-1*; *Burkewitz et al., 2020*). Calreticulin is responsible for binding and sequestering calcium in the ER lumen (*Groenendyk et al., 2021*; *Park et al., 2001*). It was previously shown that FMO protein extracted from rabbit lung can form a complex with calreticulin, suggesting a physical interaction (*Guan et al., 1991*). When testing for genetic interactions between *crt-1* and *fmo-4*, we find that *crt-1* RNAi extends WT lifespan,

as previously reported (*Burkewitz et al., 2020*), and that this lifespan extension is not additive with *fmo-4* OE (*Figure 6A*). Treating *fmo-4* KO worms with *crt-1* RNAi can significantly extend lifespan (*Figure 6—figure supplement 1A*), but this result was inconsistent (*Source data 3*). Overall, this suggests that *fmo-4* and *crt-1* are acting in the same genetic pathway. IP$_3$R, or *itr-1* in *C. elegans,* is located in the ER membrane and is critical for exporting calcium from the ER lumen into the cytosol (*Groenendyk et al., 2021*; *Berridge, 1993*). Previous results suggest that loss of *itr-1* decreases WT lifespan (*Burkewitz et al., 2020*), which our results confirm (*Figure 6B*). Importantly, we also find that *itr-1* RNAi decreases lifespan of *fmo-4* KO worms similarly (*Figure 6—figure supplement 1B*) and *fmo-4* OE (*Figure 6B*) worms to a greater extent than WT, negating the relative longevity of *fmo-4* OE worms compared to WT. Thus, *itr-1* is required to increase lifespan downstream of *fmo-4*. MCU (*mcu-1*) is located in the inner mitochondrial membrane and allows cytosolic calcium into the inner mitochondrial matrix (*Groenendyk et al., 2021*; *Marchi and Pinton, 2014*). We also find that *mcu-1* is required downstream of *fmo-4* OE, as *mcu-1* RNAi abrogates the *fmo-4* OE lifespan (*Figure 6C*). The *fmo-4* KO lifespan is slightly decreased on *mcu-1* RNAi (*Figure 6—figure supplement 1C*). These data support that *fmo-4* extends lifespan through its interactions with the calcium signaling genes, *crt-1*, *itr-1*, and *mcu-1*.

In addition to lifespan assessment, we were curious whether these genes also interact with *fmo-4* expression to modify resistance to paraquat. We exposed WT, *fmo-4* OE, and *fmo-4* KO worms to 5 mM paraquat plates with RNAi for *crt-1*, *itr-1* or *mcu-1* and assessed survival over time. Interestingly, we find that *crt-1* RNAi promotes resistance in WT worms and this effect is not additive in *fmo-4* OE worms (*Figure 6D*). Furthermore, *crt-1* RNAi extends the lifespan of *fmo-4* KO worms (*Figure 6—figure supplement 1D*), suggesting that *crt-1* and *fmo-4* act in the same pathway and that *crt-1* may act downstream of *fmo-4* to promote paraquat stress resistance. Neither *itr-1* RNAi nor *mcu-1* RNAi confers resistance to paraquat, and *fmo-4* OE resistance is lost when exposed to these RNAi (*Figure 6E, F*). The *fmo-4* KO worms show a decrease in paraquat survival when treated with *itr-1* and *mcu-1* RNAi (*Figure 6—figure supplement 1E, F*). We also measured paraquat stress resistance of *fmo-4* OE and KO worms exposed to RNAi of the voltage-dependent anion channel 1 (*vdac-1*), which is located in the outer mitochondrial membrane and also regulates calcium flow into and out of the mitochondria (*Shoshan-Barmatz et al., 2018*). Interestingly, *fmo-4* OE resistance is lost when exposed to *vdac-1* RNAi and the *fmo-4* KO worms show no difference in paraquat survival compared to the EV RNAi control (*Figure 6—figure supplement 1G, H*). These data support a model where *fmo-4* OE converges onto a shared pathway mediating paraquat stress resistance and longevity regulation. Together, we conclude that *fmo-4* interacts with ER to mitochondrial calcium signaling through *crt-1*, *itr-1*, and *mcu-1*, to promote longevity and paraquat stress resistance.

## *atf-6* KD regulates *fmo-4*-mediated longevity and paraquat resistance

Our data establish that *fmo-4* interacts with ER and mitochondrial calcium signaling to promote lifespan extension and resistance to paraquat. A previous study linked many of these components to a major regulator of UPR$^{ER}$, activating transcription factor-6 (*atf-6*) (*Burkewitz et al., 2020*). When *atf-6* is lost, lifespan is extended through changes in calcium signaling between the ER and mitochondria requiring *crt-1*, *itr-1*, and *mcu-1* (*Burkewitz et al., 2020*). Based on our transcriptomics data and *fmo-4's* interactions with calcium signaling genes, we hypothesized that *fmo-4* could interact with *atf-6* to promote longevity and paraquat resistance. To test whether *atf-6* is regulating *fmo-4*, we utilized our *fmo-4p::mCherry* transcriptional reporter strain. We measured *fmo-4* gene expression after RNAi knockdown of *atf-6* and find that *fmo-4p::mCherry* worms on *atf-6* RNAi show a consistent ~two fold increase in fluorescence (*Figure 7A, B*), similar to what we observe from calcium perturbations (*Figure 5B, C*). Since *atf-6* is one of three branches of UPR$^{ER}$, we were curious to see if *fmo-4* interacts specifically with *atf-6* or also with the other two branches, *ire-1/xbp-1* and *pek-1/atf-4* (*Read and Schröder, 2021*). We treated the *fmo-4p::mCherry* reporter worms with *ire-1*, *xbp-1*, *pek-1*, or *atf-4* RNAi and measured fluorescence. We find that knocking down the components of the IRE-1 or PEK-1 branches does not induce *fmo-4* gene expression (*Figure 7—figure supplement 1A, B*). Thus, only knockdown of the ATF-6 branch of UPR$^{ER}$ induces *fmo-4* expression.

We hypothesized that *atf-6* limits *fmo-4* expression and thus *fmo-4* acts downstream of *atf-6* knockdown to modulate lifespan extension. To test this, we assessed survival and find that while *atf-6* RNAi extends the lifespan of WT worms, as reported (*Burkewitz et al., 2020*), this effect is abrogated when

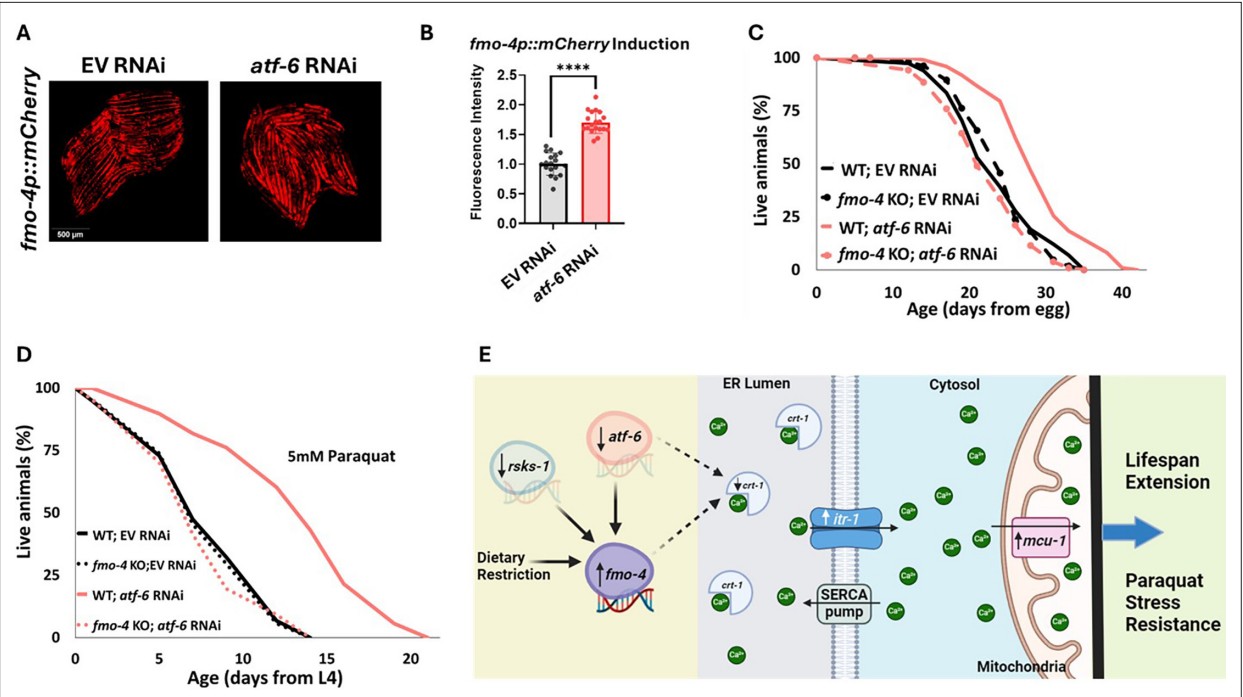

**Figure 7.** *atf-6* KD regulates *fmo-4*-mediated longevity and paraquat stress resistance. (**A**) Fluorescence imaging of the *fmo-4p::mCherry* reporter worms on empty vector (EV) and *atf-6* RNAi. (**B**) Quantification of (**A**) (n = ~20 worms per condition, three replicate experiments). (**C**) Lifespan analysis of wild-type (WT) and *fmo-4* knockout (*fmo-4* KO) worms on EV and *atf-6* RNAi (n = ~120 worms per condition, three replicate experiments). (**D**) Survival of WT and *fmo-4* KO worms on EV and *atf-6* RNAi exposed to 5 mM paraquat at L4 stage (n = ~90 worms per condition, three replicate experiments). Significance was determined at p<0.05 using log-rank analysis and significant interactions between the condition of interest and genotype was determined at p<0.01 using Cox regression analysis. (**E**) Working model showing that *fmo-4* acts downstream of multiple longevity promoters including dietary restriction, reduction in *rsks-1,* and reduction in *atf-6*. Reduced expression of *atf-6* induces *fmo-4* expression, which regulates calcium regulation from the endoplasmic reticulum (ER) to the mitochondria to promote lifespan extension and paraquat stress resistance. For imaging experiments, * denotes significant change at p<0.05 using unpaired two-tailed t test. N.S. = not significant. All replicate data can be found in *Figure 7—source data 1*; *Figure 7—source data 2*; *Figure 7—source data 3*; *Figure 7—source data 4*.

The online version of this article includes the following source data and figure supplement(s) for figure 7:

**Source data 1.** *atf-6* RNAi imaging replicates.

**Source data 2.** Quantification of *atf-6* RNAi imaging replicates.

**Source data 3.** *fmo-4* KO worms on *atf-6* RNAi lifespan replicates.

**Source data 4.** *fmo-4* KO worms on *atf-6* RNAi paraquat stress replicates.

**Figure supplement 1.** *fmo-4* gene expression is not induced by the other branches of UPR[ER].

**Figure supplement 1—source data 1.** Fluorescent images of reporter worms on *atf-4, pek-1, ire-1,* and *xbp-1* RNAi replicates.

**Figure supplement 1—source data 2.** Quantification of fluorescent images of reporter worms on *atf-4, pek-1, ire-1,* and *xbp-1* RNAi replicates.

---

*fmo-4* is knocked out (*Figure 7C*). This result suggests that *fmo-4* is indeed acting downstream of *atf-6* to promote longevity. To determine if *atf-6* is also involved in *fmo-4*-mediated paraquat resistance, we exposed WT and *fmo-4* KO worms to 5 mM paraquat plates with *atf-6* RNAi. We find that *atf-6* RNAi promotes stress resistance to paraquat, but this effect is lost when *fmo-4* is knocked out, supporting the hypothesis that *atf-6* is involved in *fmo-4*-mediated paraquat resistance (*Figure 7D*). Together, these data suggest that *fmo-4* modulates lifespan and paraquat stress resistance downstream of the reduction in *atf-6*, ultimately regulating ER to mitochondria calcium signaling (*Figure 7E*).

## Discussion

Together, our data present a model where *fmo-4* acts downstream of DR and mTOR to positively affect healthspan and longevity (*Figure 1B–C*). *fmo-4* overexpression is sufficient to provide these benefits (*Figure 3*) and does not require the DR-mediating family member, *fmo-2* (*Figure 2*). Our

results also suggest that *fmo-4* gene expression is upregulated upon changes in intracellular calcium, and that *fmo-4* interacts with calcium signaling through key genes like *crt-1, itr-1,* and *mcu-1*, to promote longevity and paraquat stress resistance (*Figures 5–6*). The relationship between *fmo-4* and calcium is further illustrated by *fmo-4* OE's susceptibility to thapsigargin, a calcium-mediated ER stress. Furthermore, we find that knocking down the UPR$^{ER}$ transcription factor and calcium regulator, *atf-6*, induces *fmo-4* gene expression and requires *fmo-4* to promote longevity (*Figure 7*). Collectively, our data suggest that *fmo-4* promotes longevity and paraquat stress resistance by regulating calcium homeostasis between the ER and mitochondria (*Figure 7E*).

Previous studies from our lab have elucidated how *C. elegans fmo-2* promotes longevity through endogenous metabolism (*Choi et al., 2023*). Based on the conservation within the *fmo* gene family and that multiple Fmos are induced in long-lived worms and mammals, it was reasonable to hypothesize that Fmos could play overlapping roles in longevity regulation. Our data here support the broader hypothesis that Fmos have a conserved role in aging, but interestingly, that the mechanisms by which they modulate aging are separable. Importantly, we find that *fmo-4* is required for multiple longevity-promoting pathways, including DR and mTOR, but is not required for lifespan extension through the hypoxic response, insulin-like signaling, or cytochrome c reductase pathways. We also find that *fmo-4* does not require *fmo-2,* but that *fmo-2* does require *fmo-4*. This is interesting because not only do these data tell us that *fmo-4* is important in the context of longevity, but also that it acts distinctly from *fmo-2*.

After creating ubiquitous and hypodermal-specific *fmo-4* overexpressing strains, we find that *fmo-4* is sufficient for longevity and paraquat stress resistance but not heat, tunicamycin, or thapsigargin stress. These results further differentiate *fmo-4* from *fmo-2* and also provide insight into *fmo-4's* potential roles in the cell. For instance, *fmo-4* OE worms' resistance to paraquat could suggest a role in responding to or controlling reactive oxygen species (ROS) production in the mitochondria. Additionally, *fmo-4* OE worms' sensitivity to thapsigargin likely points towards an involvement in ER calcium regulation. These are both interesting plausible roles for *fmo-4,* and may go hand in hand, as calcium levels are known to impact ROS levels (*Görlach et al., 2015*), and changes in ROS levels are known to impact ER calcium release (*Görlach et al., 2015*). Based on this and the data with calcium perturbations, we speculate that FMO-4 acts as a calcium sensor, and when calcium levels are too high or too low, FMO-4 responds by regulating downstream calcium signaling proteins to restore homeostasis in the cell. As FMO-4 is a predicted ER transmembrane protein, we hypothesize that FMO-4 is sensing changes in calcium levels in the ER and/or cytosol.

We note that our data reveal genetic interactions between *fmo-4* and other longevity pathways like DR, mTOR, FMO-2, and ATF-6. However, it is important to also understand on the protein level how FMO-4 is interacting with these pathways and how FMO-4 is regulating downstream calcium signaling components to promote longevity and paraquat stress resistance. For instance, it was previously shown that rabbit FMO protein forms a complex with calreticulin (*Guan et al., 1991*). It is possible that an increase in FMO-4 leads to more binding between FMO-4 and calreticulin, which ultimately prevents calreticulin from binding ER calcium. This would then allow for proper calcium flux from the ER to the mitochondria via the IP$_3$R and MCU, respectively. Our future work will look into these protein interactions so that we can further tease apart the FMO-4-mediated longevity pathway. We also note that we have utilized RNAi knockdown rather than knockout mutants to assess these genetic interactions with *fmo-4*. In particular, we were wary of knocking out crucial calcium regulating genes, like *itr-1* and *mcu-1,* that already result in some level of sickness in the worms when knocked down (*Figure 6B*) or could potentially lead to other confounding metabolic changes if knocked out. We were able to obtain robust and reproducible results using RNAi knockdown, but recognize the caveats that come with not testing full deletion mutants.

While our transcriptomics analysis and calcium manipulations suggest *fmo-4* involvement in calcium regulation, there are other measures that can better assess an interaction between *fmo-4* and calcium. Carbachol and EDTA supplementation are not perfect assessments of altered calcium levels, as carbachol does not act in every tissue and EDTA may be depleting more than just calcium from the cell. Thus, future studies should measure calcium levels and calcium flux in *fmo-4* OE and KO worms using GCaMP expressing worms to determine the direct link between *fmo-4* expression and calcium in the cell. Additionally, to determine if FMO-4 is acting as a calcium sensor or interacting with one to regulate calcium homeostasis, development of FMO-4 antibodies for immunoprecipitation

assays would be useful (*Kaufmann and Sauter, 2017*). Together, it will be important to establish the biochemical parameters of the FMO-4 calcium interaction.

As calcium signaling occurs between the ER and the mitochondria, our results could have interesting implications in ER-mitochondrial metabolism. We would expect that FMO-4 regulates ER-mitochondria metabolism because (1) FMO-4 is a predicted ER transmembrane protein and blocking the expression of another ER transmembrane protein, ATF-6, induces the expression of *fmo-4*, (2) *fmo-4* OE worms are highly sensitive to ER calcium stress but resistant to ROS induced stress, (3) lifespan extension driven by *fmo-4* depends on mitochondrial calcium import (*mcu-1*), and (4) Fmo gene expression is induced by mitochondrial inhibitors (*Huang et al., 2021*). Additionally, changes in mitochondrial calcium import affect various mitochondrial measures like respiration, dynamics (i.e. fission and fusion), membrane potential, and TCA cycle activity (*Duchen, 2000*). Future studies will delve into how *fmo-4* perturbations impact each of these measures and if they act in the *fmo-4*-mediated longevity pathway. Based on our data, we speculate that *fmo-4* OE may increase mitochondrial calcium by regulating *mcu-1* activity, and that *fmo-4* OE worms may in turn be regulating ROS production in the mitochondria. We hypothesize that changes in *fmo-4* expression will alter mitochondrial respiration, dynamics, membrane potential, and TCA cycle activity. Taken together, future experiments will shed light on the role that FMOs play in calcium regulation and mitochondrial metabolism and will help to further tease apart the FMO-mediated longevity pathway.

# Materials and methods
## Strains and growth conditions
Standard *C. elegans* cultivation procedures were used as previously described (*Leiser et al., 2015*). Briefly, all worm strains were maintained on solid nematode growth media (NGM) using *E. coli* OP50 throughout life except where double stranded (ds) RNAi (*E. coli* HT115) were used. Worms were transferred using a platinum wire. All worm strains were kept at 20 °C. RNAi used is listed in *Supplementary file 3*. Worm strains are listed in *Supplementary file 4*. Genotyping primers are listed in *Supplementary file 5*.

## Development assays
Animals were synchronized by placing 10 gravid adult worms on NGM plates seeded with *E. coli* OP50 to lay eggs for 1 hr at 20 °C. The gravid adult worms were then removed, and the eggs were allowed to hatch and develop at 20 °C until larval stage 2 (L2). At this point the L2 worms were moved to individual 35 mm NGM plates seeded with *E. coli* OP50, one worm per plate. This was done for each worm strain tested and ten total 35 mm plates per worm strain were prepared. The worms were then followed through development and watched hourly during young adulthood to score the time of the first egg lay, as previously described (*Leiser et al., 2015*). Development time was reported in hours since egg lay. Three replicate experiments were performed. For development on DMSO control or thapsigargin, the same protocol was followed except that the L2 worms were moved to individual 35 mm NGM plates seeded with *E. coli* OP50 that were spotted with either DMSO or 1 mg/mL thapsigargin, one worm per plate.

## Stress resistance assays
### Paraquat stress assay
Paraquat (Methyl viologen dichloride hydrate, 856177, Sigma-Aldrich) was used to induce oxidative stress. Worms were synchronized from eggs on either NGM plates seeded with *E. coli* OP50 or RNAi plates seeded with HT115 strain expressing dsRNAi for a particular gene. At L4 stage, 30 worms were transferred to either NGM plates or RNAi-FUdR (40690016, Bioworld) plates containing 5 mM paraquat dissolved in water. Three plates per strain per condition were prepared, for a total of 90 worms per condition. As previously described, worms were then scored every other day and considered dead when they did not move in response to prodding under a dissection microscope (*Leiser et al., 2015*). Worms that crawled off the plate were not considered, but ruptured worms were noted and considered. Three replicate experiments were performed. Log-rank test was used to derive p-value for paraquat stress resistance survival assays using $p < 0.05$ cut-off threshold compared to EV or wild-type

controls. Cox regression was also used to assess interactions between genotype and condition for paraquat stress resistance survival assays using $p < 0.01$ cut-off threshold compared to controls.

### Heat stress assay

Worms were synchronized from eggs on NGM plates seeded with *E. coli* OP50. Once the worms reached day 1 of adulthood, 25 worms of each strain were transferred to 35 mm NGM plates seeded with *E. coli* OP50 (*Burkewitz et al., 2020*). Four plates per strain per condition were prepared, for a total of 100 worms per condition. The plates were placed in a single layer on a tray and incubated at 37 °C for 3 hr. After 3 hr, the plates were removed and placed at 20 °C for 24 hr to give the worms time to recover. After 24 hr, the number of dead and alive worms was counted. % Alive was calculated as (# alive/# total)×100 and graphed in Graphpad Prism using unpaired two-tailed t tests with Welch's correction as well as one-way ANOVA to determine significance. Three replicate experiments were performed.

### Tunicamycin stress assay

*E. coli* OP50 was cultured at 37 °C shaking overnight and then centrifuged and concentrated to 100 x. 100 µL of bacteria was added to 9.9 mL of S-media to create a 1 x mixture. 100 µL of this bacteria/S--media mixture was added to the wells of a Falcon non tissue culture treated clear flat bottom 96 well plate (Falcon, 351172). Approximately 20 eggs per worm strain were added to the wells of the 96 well plate as previously described (*Leiser et al., 2015*). Three replicates per worm strain were added to the 96-well plate, for a total of 60 eggs per worm strain. Then either 0, 1, 2, or 5 µg/mL of tunicamycin (Sigma, T7765) diluted in dimethyl sulfoxide (DMSO) (Sigma, D2650) was added to the wells. Total volume per well was 100 µL. The plate was incubated covered at room temperature on an orbital shaker for 72 hr. After 72 hr, each well containing worms was rinsed with 100 µL M9 and then added to individual 35 mm NGM plates seeded with *E. coli* OP50 to sit covered at room temperature overnight. The following day, plates were assessed for live adults. % Alive Adults was calculated by determining how many of the three technical replicates of a given strain showed at least one live adult and then graphed in Microsoft Excel. Three replicate experiments were performed.

### Thapsigargin stress assay

To assess development of worms during chronic ER calcium stress, NGM plates seeded with *E. coli* OP50 were spotted with 25 µL 1 mg/mL thapsigargin in DMSO (Sigma) or DMSO only directly on to the *E. coli* OP50 lawns and allowed to dry overnight. Then, 30 synchronized L1's per worm strain were added directly to the spotted lawns (*Burkewitz et al., 2020*). Forty-eight hours later, worms were picked off these plates, added to a 2% agarose pad on a glass slide, anesthetized in 0.5 M sodium azide (Sigma), and imaged at 6.3 x magnification (brightfield) with the LASx software and Leica scope. Three replicates were performed. Each worm was recorded in ImageJ. Data were analyzed in GraphPad Prism using unpaired two-tailed t tests with Welch's correction.

## Thrashing assays

Animals were synchronized by placing 10 gravid adult worms on NGM plates seeded with *E. coli* OP50 and allowing them to lay eggs for 2 hr at 20 °C. The gravid adult worms were then removed, and the eggs were allowed to hatch and develop at 20 °C until day 1 adulthood. Worms were placed in a drop of M9 solution, as previously described (*Leiser et al., 2015*). The individual body bends were counted at maximum rate for 30 s. Thrashing was assayed on day 2 and day 10 of adulthood. The animals that were not used for the day 2 assay were transferred to fresh fed FUdR plates four times until they were ready to be assayed. Three replicates were performed. Data were analyzed in GraphPad Prism using unpaired two-tailed t tests with Welch's correction as well as one-way ANOVA.

## Lifespan assays

Gravid adults were placed on NGM plates containing 1 mM β-D-isothiogalactopyranoside (IPTG), 25 µg/ml carbenicillin, and the corresponding RNA interference (RNAi) clone from the Vidal or Ahringer RNAi library. 200 µL of HT115 bacteria expressing double stranded (ds) RNA of either the control empty vector (EV) or the RNAi of interest at optical density (OD) of 3.0 and concentration of 3 X was added to each plate. After 3 hr, the adults were removed, and the eggs were allowed to

develop at 20 °C until they reached late L4/young adult stage. From here, 70 worms were placed on each RNAi plate and transferred to fresh RNAi +FUDR plates on day 1, day 2, day 4, and day 6 of adulthood. A minimum of two plates per strain per condition were used per replicate experiment. Experimental animals were scored every 2–3 d and considered dead when they did not move in response to prodding under a dissection microscope. Worms that crawled off the plate were not considered, but ruptured worms were considered as previously described (*Leiser et al., 2015*). A similar method was used for non-RNAi lifespan experiments, except the plates did not contain IPTG and worms were fed *E. coli* OP50. 200 μL of *E. coli* OP50 at OD of 3.0 and concentration of 3 X was added to each plate. A similar method was also used for carbachol (Thermo Fisher, L06674.14) and EDTA (Thermo Fisher, AM9260G) supplementation lifespan experiments, except 50 μM carbachol or 50 μM EDTA was added to the NGM plates without IPTG, and worms were fed *E. coli* OP50. 200 μL of *E. coli* OP50 at OD of 3.0 and concentration of 3 X was added to each plate. Optimal concentrations of carbachol and EDTA (50 μM) were determined by assessing survivability of worms exposed to a range of concentrations. Log-rank test was used to derive p-value for lifespan assays using $p < 0.05$ cut-off threshold compared to EV or wild-type controls. Cox regression was also used to assess interactions between genotype and condition for lifespan assays sing $p < 0.01$ cut-off threshold compared to controls.

## Dietary restriction (sDR) lifespan assay

Gravid adults were placed on NGM plates seeded with 200 μL *E. coli* OP50. After 3 hr, the adults were removed, and the eggs were allowed to develop at 20 °C until they reached late L4/young adult stage. From here, 70 worms were transferred to fed FUdR plates seeded with 200 μL of *E. coli* OP50 at OD of 3.0 and concentrated 3 X on days 1 and 2 of adulthood. On day 3 of adulthood, DR conditions were transferred to plates with $10^9$ seeded lawns and transferred every other day four times while the corresponding controls were transferred equally to fed ad-lib plates. This form of DR is termed solid DR (sDR; *Greer and Brunet, 2009*). 80 μL of 10 mM palmitic acid (Sigma-Aldrich) dissolved in 100% EtOH was added to the outer rim of the plate to prevent fleeing. A minimum of two plates per strain per condition were used per replicate experiment. Experimental animals were scored every 2–3 d and considered dead when they did not move in response to prodding under a dissection microscope. Worms that crawled off the plate were not considered, but ruptured worms were considered as previously described (*Leiser et al., 2015*).

## Transcriptomic analysis

Approximately 600 day 1 adult worms per biological replicate were washed with M9 buffer three times, frozen in liquid nitrogen, and then stored at –80 °C. RNA was extracted by adding 500 mL of Trizol reagent to the frozen worm pellets, followed by three freeze-thaw cycles with liquid nitrogen and water bath at 42 °C. Then, 500 μL of ethanol was added to the samples, and RNA was isolated using the Direct-Zol Miniprep Plus Kit (Cat#R2072). Purified RNA was sent to Novogene (Novogene Corporation Inc) for sequencing on the Illumina HWI-ST1276 instrument. Messenger RNA was purified from total RNA by using poly-T oligo-attached magnetic beads. After fragmentation, first strand cDNA was synthesized using random hexamer primers, followed by second strand cDNA. The library was constructed, which entailed end repair, A-tailing, adapter ligation, size selection, amplification, and purification, and then was sequenced on an Illumina device using paired-end sequencing. Gene ontology analysis was done using the National Institutes of Health DAVID Bioinformatics tool. The dataset has been submitted to GEO NCBI (GEO accession #GSE288007).

## Gene expression assays

RNA was isolated from day 1 adult worms (approximately 600 worms per strain) following three rounds of freeze-thaw in liquid nitrogen using Invitrogen's Trizol extraction method, similar to the method described above. 1 μg of isolated and purified RNA was reverse transcribed to cDNA using SuperScript II Reverse Transcriptase (18064071, Invitrogen,). Gene expression levels were measured using 600 ng of cDNA and SYBR Green PCR Mastermix (A25742, Applied Biosystems) with primers at 10 μM concentration. mRNA levels were normalized using Y45F10D.4 as a reference gene (*Taki and Zhang, 2013*). List of qPCR primers used are in *Supplementary file 6*.

### *fmo-4* induction on RNAi

Gravid *fmo-4p::mCherry* transcriptional reporter adult animals were placed on NGM plates containing 1 mM β-D-isothiogalactopyranoside (IPTG), 25 μg/ml carbenicillin, and the corresponding RNAi clone from the Vidal or Ahringer RNAi library. After 3 hr, the adults were removed, and the eggs were allowed to develop at 20 °C until they reached late L4/young adult stage. Then 40–50 worms per plate were transferred to similar plates that contain FUdR for overnight. The following day, ~20 worms per condition were picked off these plates, added to a 2% agarose pad on a glass slide, anesthetized in 0.5 M sodium azide (Sigma), and imaged at 6.3 x magnification with the LASx software and Leica scope using the mCherry fluorescence channel (*Miller et al., 2022*). Three replicates were performed. Fluorescent intensity in the mCherry channel was measured in ImageJ. Brightness of images within a dataset was increased to the same level. Data were analyzed in GraphPad Prism using unpaired two-tailed t tests with Welch's correction.

### *fmo-4* induction on carbachol

Gravid *fmo-4p::mCherry* transcriptional reporter adult animals were placed on NGM plates seeded with *E. coli* OP50. After 3 hr, the adults were removed, and the eggs were allowed to develop at 20 °C until they reached late L4/young adult stage. Then 30 worms were transferred to NGM plates seeded with *E. coli* OP50 also containing either 300 μM carbachol (Thermo Fisher, L06674.14) or water. An optimal concentration of carbachol was determined by first assessing survivability of worms exposed to a range of concentrations and then by assessing fluorescence of the *fmo-4p::mCherry* reporter worms exposed to a range of concentrations. The worms were incubated for 24 hr at 20 °C. After 24 hr, ~20 worms per condition were then picked off these plates and added to unseeded NGM plates, anesthetized in 0.5 M sodium azide (Sigma), and imaged at 6.3 x magnification with the LASx software and Leica scope using the mCherry fluorescence channel (*Miller et al., 2022*). Three replicates were performed. Each worm was measured for fluorescence in ImageJ. Data were analyzed in GraphPad Prism using t tests.

### *fmo-4* induction assay on EDTA

Gravid *fmo-4p::mCherry* transcriptional reporter adult animals were placed on NGM plates seeded with *E. coli* OP50. After 3 hr, the adults were removed, and the eggs were allowed to develop at 20 °C until they reached late L4/young adult stage. Then 30 worms were transferred to NGM plates seeded with *E. coli* OP50 topically spotted with 150 μL of 0.5 M ethylenediaminetetraacetic acid (EDTA) (ThermoFisher, AM9260G) or 150 μL of water. An optimal concentration of EDTA was determined by first assessing survivability of worms exposed to a range of concentrations and then by assessing the fluorescence of the *fmo-4p::mCherry* reporter worms exposed to a range of concentrations. The worms were incubated for 24 hr at 20 °C. After 24 hr, ~20 worms per condition were then picked off these plates and added to unseeded NGM plates, anesthetized in 0.5 M sodium azide (Sigma), and imaged at 6.3 x magnification with the LASx software and Leica scope using the mCherry fluorescence channel (*Miller et al., 2022*). Three replicates were performed. Each worm was recorded in ImageJ. Data were analyzed in GraphPad Prism using t tests.

### GCaMP7f induction assay on carbachol and EDTA

Gravid GCaMP7f (SWF702; *Dag et al., 2023*) adult worms were placed on NGM plates seeded with *E. coli* OP50. After 3 hr, the adults were removed, and the eggs were allowed to develop at 20 °C until they reached late L4/young adult stage. Then 30 worms were transferred to NGM plates seeded with *E. coli* OP50 topically spotted with 150 μL of 0.5 M EDTA(Thermo Fisher, AM9260G), 300 μM carbachol (Thermo Fisher, L06674.14), or water. The worms were incubated for 24 hr at 20 °C. After 24 hr, ~20 worms per condition were then picked off these plates and added to unseeded NGM plates, anesthetized in 0.5 M sodium azide (Sigma), and imaged at 6.3 x magnification with the LASx software and Leice scope using the GFP fluorescence channel. Three replicates were performed. Each worm was recorded in ImageJ, with focus on the head region to assess neuronal expression. Data were analyzed in GraphPad Prism using t tests. For time course assays, gravid GCaMP7f adult worms were placed on unseeded NGM plates, anesthetized in 0.5 M sodium azide (Sigma), and imaged at

6.3 x magnification with the LASx software and Leica scope using the GFP fluorescence channel. Then 2 µL of either 300 µM carbachol or 10 mM EDTA were added to the worms and they were imaged using the GFP fluorescence channel at 0.5 min, 5 min, 10 min, 15 min, and 20 min. Three replicates were performed. Each worm was recorded in ImageJ, with focus on the head region to assess neuronal expression. Data were analyzed in GraphPad Prism using t tests.

## Statistical analyses

Log-rank test was used to derive p-value for lifespan and stress resistance survival assays using $p<0.05$ cut-off threshold compared to EV or wild-type controls. Cox regression was also used to assess interactions between genotype and condition for lifespans and stress resistance survival assays using $p<0.01$ cut-off threshold compared to controls. *Source data 3* provides the results of the Log-rank test and Cox regression analysis, which were run in RStudio.

## Acknowledgements

This work was supported by grants from NIH. AMT was supported by NIH T32AG000114 and the University of Michigan Rackham Research Grant. SFL was supported by R01AG075061 and the Glenn Foundation for Medical Research.

## Additional information

### Competing interests

Scott F Leiser: Reviewing editor, eLife. The other authors declare that no competing interests exist.

### Funding

| Funder | Grant reference number | Author |
| --- | --- | --- |
| National Institutes of Health | R01AG075061 | Scott F Leiser |
| National Institutes of Health | T32AG000114 | Angela M Tuckowski |
| Glenn Foundation for Medical Research | | Scott F Leiser |
| University of Michigan | Rackham Research Grant | Angela M Tuckowski |

The funders had no role in study design, data collection and interpretation, or the decision to submit the work for publication.

### Author contributions

Angela M Tuckowski, Conceptualization, Formal analysis, Validation, Investigation, Visualization, Methodology, Writing - original draft, Writing – review and editing; Safa Beydoun, Ajay Bhat, Aditya Sridhar, Mira Bhandari, Investigation, Writing – review and editing; Elizabeth S Kitto, Formal analysis, Investigation, Writing – review and editing; Marshall B Howington, Conceptualization, Investigation, Writing – review and editing; Kelly Chambers, Investigation; Scott F Leiser, Conceptualization, Resources, Supervision, Funding acquisition, Project administration, Writing – review and editing

### Author ORCIDs

Angela M Tuckowski (ID) https://orcid.org/0000-0002-9125-4780
Scott F Leiser (ID) https://orcid.org/0000-0002-8003-2955

Reviewer #1 (Public review): https://doi.org/10.7554/eLife.99971.3.sa1
Reviewer #2 (Public review): https://doi.org/10.7554/eLife.99971.3.sa2
Reviewer #3 (Public review): https://doi.org/10.7554/eLife.99971.3.sa3
Author response https://doi.org/10.7554/eLife.99971.3.sa4

# Additional files

## Supplementary files

Supplementary file 1. qPCR validation of worm strains. All replicate data can be found in *Supplementary file 7*.

Supplementary file 2. RNA-sequencing calcium-related transcripts from DAVID analysis.

Supplementary file 3. List of RNAi used.

Supplementary file 4. List of worm strains used in this paper.

Supplementary file 5. List of genotyping primers used to validate worm strains.

Supplementary file 6. List of qPCR primers used to validate worm strains.

Supplementary file 7. qPCR validation of worm strains replicates.

MDAR checklist

Source data 1. RNA-sequencing results showing the significantly upregulated and significantly downregulated transcripts when *fmo-4* is overexpressed in *C. elegans*.

Source data 2. DAVID analysis shows that calcium ion binding transcripts are regulated when *fmo-4* is overexpressed in *C. elegans*.

Source data 3. Log-rank and Cox regression analyses of all lifespans and stress resistance assays which were run in RStudio.

## Data availability

Replicate data for all figures and figure supplements can be found in the source data files. Additionally, the RNA-sequencing dataset has been submitted to GEO NCBI (GEO accession #GSE288007).

The following dataset was generated:

| Author(s) | Year | Dataset title | Dataset URL | Database and Identifier |
|---|---|---|---|---|
| Tuckowski AM, Beydoun S, Kitto ES, Bhat A, Howington MB, Sridhar A, Bhandari M, Chambers K, Leiser SF | 2025 | fmo-4 promotes longevity and stress resistance via ER to mitochondria calcium regulation in *C. elegans* | https://www.ncbi.nlm.nih.gov/geo/query/acc.cgi?acc=GSE288007 | NCBI Gene Expression Omnibus, GSE288007 |

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
