## [Editor Report · eLife Assessment]

This **important** study offers **convincing** evidence that *fmo-4* plays essential roles in established lifespan interventions and downstream of its paralog *fmo-2*. The work is of substantial benefit for our understanding of this enzyme family, underscoring their importance in longevity and stress resistance. The study also suggests a connection between *fmo-4* and dysregulation of calcium signalling, with conclusions and interpretations based on **solid** genetic methodology and evidence.

---

## [Referee Report · Reviewer #1 (Public review)]

Summary:

This interesting and well-written article by Tuckowski et al. summarizes work connecting the flavin-containing monooxygenase FMO-4 with increased lifespan through a mechanism involving calcium signaling in the nematode *Caenorhabditis elegans*.

The authors have previously studied another fmo in worms, FMO-2, prompting them to look at additional members of this family of proteins. They show that fmo-4 is up in dietary restricted worms and necessary for the increased lifespan of these animals as well as of rsks-1 (s6 kinase) knockdown animals. They then show that overexpression of fmo-4 is sufficient to significantly increase lifespan, as well as healthspan and paraquat resistance. Further, they demonstrate that overexpression of fmo-4 solely in the hypodermis of the animal recapitulates the entire effect of fmo-4 OE.

In terms of interactions between fmo-2 and fmo-4 they show that fmo-4 is necessary for the previously reported effects of fmo-2 on lifespan, while the effects of fmo-4 do not depend on fmo-2.

Next the authors use RNASeq to compare fmo-4 OE animals to wild type. Their analyses suggested the possibility that FMO-4 was modulating calcium signaling, and through additional experiments specifically identified the calcium signaling genes crt-1, itr-1, and mcu-1 as important fmo-4 interactors in this context. As previously published work has shown that loss of the worm transcription factor atf-6 can extend lifespan through crt-1, itr-1 and mcu-1, the authors asked about interactions between fmo-4 and atf-6. They showed that fmo-4 is necessary for both lifespan extension and increased paraquat resistance upon RNAi knockdown of atf-6.

Overall this clearly written manuscript summarizes interesting and novel findings of great interest in the biology of aging, and suggests promising avenues for future work in this area.

Strengths:

This paper contains a large number of careful, well executed and analysed experiments in support of its existing conclusions, and which also point toward significant future directions for this work. In addition it is clear and very well written.

Weaknesses:

Within the scope of the current work there are no major weaknesses. That said, the authors themselves note pressing questions beyond the scope of this study that remain unanswered. For instance, the mechanistic nature of the interactions between FMO-4 and the other players in this story, for example in terms of direct protein-protein interactions, is not at all understood yet. Further, powerful tools such as GCaMP expressing animals will enable a much more detailed understanding of what exactly is happening to calcium levels, and where and when it is happening, in these animals.

---

## [Referee Report · Reviewer #2 (Public review)]

Summary:

Members of a conserved family of flavin-containing monooxygenases (FMOs) play key roles in lifespan extension induced by diet restriction and hypoxia. In *C. elegans*, fmo-2 has received the majority of attention, but there are multiple fmo genes in both worms and mammals, and how overlapping or distinct the functional roles of these paralogs are remains unclear. Here Tuckowski et al. identify that a new family member, fmo-4, is also a positive modulator of lifespan. Based on differential requirements of fmo-2 and fmo-4 in stress resistance and lifespan extension paradigms, however, the authors conclude that fmo-4 acts through mechanisms that are distinct from fmo-2. Ultimately, the authors place fmo-2 genetically within a pathway involving atf-6, calreticulin, the IP3 receptor, and mitochondrial calcium uniporter, which was previously shown to link ER calcium homeostasis to mitochondrial homeostasis and longevity. The authors thus achieve their overarching aim to reveal that different FMO family members regulate stress resistance and lifespan through distinct mechanisms. Furthermore, because the known enzymatic activity of FMOs involves oxygenating xenobiotic and endogenous metabolites, these findings highlight a potential new link between redox/metabolic homeostasis and ER-mitochondrial calcium signaling.

Strengths:

The authors demonstrate links between multiple conserved life-extending signaling pathways and fmo-4, expanding both the significance and mechanistic diversity of FMO-family genes in aging and stress biology.

The authors use genetics to discover an interesting and unanticipated new link between FMOs and calcium pathways known to regulate lifespan.

The genetic epistasis patterns for lifespan and stress resistance phenotypes are generally clean and compelling.

Weaknesses:

The authors achieve a necessary and valuable first step with regard to linking FMO-4 to calcium homeostasis, but the mechanisms involved remain preliminary at this stage. Specifically, the genetic interactions between fmo-4 and conserved mediators of calcium transport and signaling are convincing, but a putative molecular mechanism by which the activity of FMO-4 would alter subcellular calcium transport remains unclear and potentially indirect. The authors effectively highlight this gap as a key pursuit for subsequent studies.

The authors have shown that carbachol and EDTA produce the expected effects on a cytosolic calcium reporter in neurons, supporting the utility of the chemical approach in general, but validating that carbachol, EDTA and fmo-4 itself have an impact on calcium in the tissues and subcellular compartments relevant to the lifespan phenotypes would still be valuable in supporting the overall model. Notably, however, the hypodermal-specific role of FMO-4 suggests potential cell non-autonomous regulation of lifespan, such that this pathway may ultimately involve complex inter-cellular signaling that would necessitate substantially more time and effort.

Employing mutants and more sophisticated genetic tools for modulating calcium transport or signaling (in addition to RNAi) would strengthen key conclusions and/or help to elucidate tissue- or age-specific aspects of the proposed mechanism.

---

## [Referee Report · Reviewer #3 (Public review)]

Summary:

The authors assessed the potential involvement of fmo-4 in a diverse set of longevity interventions, showing that this gene is required for DR and S6 kinase knockdown related lifespan extension. Using comprehensive epistasis experiments they find this gene to be a required downstream player in the longevity and stress resistance provided by fmo-2 overexpression. They further showed that fmo-4 ubiquitous overexpression is sufficient to provide longevity and paraquat (mitochondrial) stress resistance, and that overexpression specifically in the hypodermis is sufficient to recapitulate most of these effects.

Interestingly, they find that fmo-4 overexpression sensitizes worms to thapsigargin during development, an effect that they link with a potential dysregulation in calcium signalling. They go on to show that fmo-4 expression is sensitive to drugs that both increase or decrease calcium levels, and these drugs differentially affect lifespan of fmo-4 mutants compared to wild-type worms. Similarly, knockdown of genes involved in calcium binding and signalling also differentially affect lifespan and paraquat resistance of fmo-4 mutants.

Finally, they suggest that atf-6 limits the expression of fmo-4, and that fmo-4 is also acting downstream of benefits produced by atf-6 knockdown.

Strengths:

• comprehensive lifespans experiments: clear placement of fmo-4 within established longevity interventions.

• clear distinction in functions and epistatic interactions between fmo-2 and fmo-4 which lays a strong foundation for a longevity pathway regulated by this enzyme family.

Weaknesses:

• no obvious transcriptomic evidence supporting a link between fmo-4 and calcium signalling: either for knockout worms or fmo-4 overexpressing strains.

• no direct measures of alterations in calcium flux, signalling or binding that strongly support a connection with fmo-4.

• no measures of mitochondrial morphology or activity that strongly support a connection with fmo-4.

• lack of a complete model that places fmo-4 function downstream of DR and mTOR signalling (first Results section), fmo-2 (second Results section) and at the same time explains connection with calcium signalling.

Comments on revisions:

The authors have addressed and fixed all the private comments we had made. In terms of the public comments, I think nothing has changed in terms of strengths and weaknesses. They have multiple independent results (drugs, RNAi and transcriptomics) that suggest a connection between fmo-4 and calcium regulation, but there is no strong evidence for what this connection is. The work still lacks direct measures of calcium, ER or mitochondrial function in relation to fmo-4 (which they acknowledge in the discussion). The first four sections strongly place fmo-4 within established longevity interventions, but their model doesn't explain how calcium regulation would fit into these.

---

## [Author Response]

The following is the authors’ response to the original reviews.

**Public Reviews:**

**Reviewer 1:**
Comment 1: Within the scope of the current work there are no major weaknesses. That said, the authors themselves note pressing questions beyond the scope of this study that remain unanswered. For instance, the mechanistic nature of the interactions between FMO-4 and the other players in this story, for example in terms of direct protein-protein interactions, is not at all understood yet.

We thank the reviewer for the positive review, and fully agree and acknowledge that there are unanswered questions for future studies that are beyond the scope of this manuscript.

**Reviewer 2:**
Comment 1: The effects of carbachol and EDTA on intracellular calcium levels are inferred, especially in the tissues where fmo-4 is acting. Validating that these agents and fmo-4 itself have an impact on calcium in relevant subcellular compartments is important to support conclusions on how fmo-4 regulates and responds to calcium.

We thank the reviewer for this important suggestion. We agree that carbachol and EDTA can be broad agents and validating that they are altering calcium levels is very useful. While this is technically challenging, we attempted to address this by using neuronally expressed GCaMP7f calcium indicator worms and measuring their GFP fluorescence upon exposure to carbachol and EDTA. Assessing both short term and long term exposure to these agents, we were able to show that carbachol increases GFP fluorescence, indicating an increase in calcium levels, and EDTA decreases GFP fluorescence, indicating a decrease in calcium levels. Unfortunately, because FMO-4 is not neuronally expressed, we were not able to test the effects of FMO-4 on calcium in this strain, which would require hypodermal expression and possibly short-term modification of *fmo-4* expression to test. We have made sure to temper our language about the indirect measures we used.

Comment 2: Experiments are generally reliant on RNAi. While in most cases experiments reveal positive results, indicating RNAi efficacy, key conclusions could be strengthened with the incorporation of mutants.

We appreciate and value this suggestion and agree that mutants could be helpful to strengthen our conclusions. We address this caveat in the discussion of the revised manuscript. We explain that we were concerned about knocking out key calcium regulating genes like *itr-1* and *mcu-1* that either already result in some level of sickness in the worms when knocked down (*itr-1*) or could lead to confounding metabolic changes if knocked out. We do find that our RNAi lifespan results are robust and reproducible, but we also understand and recognize the caveats that come with using RNAi knockdown instead of full deletion mutants.

**Reviewer 3:**
Comment 1: no obvious transcriptomic evidence supporting a link between fmo-4 and calcium signaling: either for knockout worms or fmo-4 overexpressing strains.

We thank the reviewer for this feedback. While there is some transcriptomic evidence, we agree that it is not overwhelming evidence. We do think that this evidence, combined with the phenotype observed under thapsigargin (i.e., significant reduction in worm size and significant delay or prevention of development), in addition to the genetic connections to calcium regulation, provide additional compelling evidence that FMO-4 interacts with calcium signaling.

Comment 2: no direct measures of alterations in calcium flux, signalling or binding that strongly support a connection with fmo-4.

As described in reviewer 2 comment 1, we have successfully used GCaMP7f worms to assess calcium flux upon exposure to carbachol and EDTA. This approach confirmed the changes in calcium expected from these compounds. Unfortunately, because FMO-4 is not neuronally expressed, we were not able to test the effects of FMO-4 on calcium in this strain, which would require hypodermal expression and possibly short-term modification of *fmo-4* expression to test. We have made sure to temper our language about the indirect measures we used.

Comment 3: no measures of mitochondrial morphology or activity that strongly support a connection with fmo-4.

This is a great point, and something we are currently working on to include for a future manuscript.

Comment 4: lack of a complete model that places fmo-4 function downstream of DR and mTOR signalling (first Results section), fmo-2 (second Results section) and at the same time explains connection with calcium signalling.

We thank the reviewer for this helpful feedback. We have included a more complete working model in our revision.

**Recommendations for the authors:**

**Reviewer 1:**
Comment 1: "We utilized fmo-4 (ok294) knockout (KO) animals on five conditions reported to extend lifespan in *C. elegans*." Here I believe "fmo-4 (ok294)" should be "fmo-4(ok294)". (No space).

We thank the reviewer for this helpful revision. We have made this change as suggested.

Comment 2: "Wild-type (WT) worms on DR experience a ~35% lifespan extension compared to fed WT worms, but when fmo-4 is knocked out this extension is reduced to ~10% and this interaction is significant by cox regression (p-value < 4.50e-6)." Here "cox regression" should be "Cox regression".

We have made this change as suggested.

Comment 3: "Having established this role, we continued lifespan analyses of fmo-4 KO worms exposed to RNAi knockdown of the S6-kinase gene rsks-1 (mTOR signaling), the von hippel lindau gene vhl-1 (hypoxic signaling), the insulin receptor daf-2 (insulin-like signaling), and the cytochrome c reductase gene cyc-1 (mitochondrial electron transport chain, cytochrome c reductase) (Fig 1C-F)." Here "von hippel lindau" should be "Von Hippel-Lindau".

We have made this change as suggested.

Comment 4: In three instances in the caption of Figure 5, the "4" in fmo-4 is not italicized when it should be.

We have made this change as suggested.

Comment 5: In two instances in the caption of Figure 7, the "4" in fmo-4 is not italicized when it should be, and in one instance in the caption of Figure 7, the "6" in atf-6 is not italicized when it should be.

We have made this change as suggested.

Comment 6: "Supplemental Data 3 provides the results of the Log-rank test and Cox regression analysis, which were run in Rstudio." Here Rstudio should be RStudio.

We have made this change as suggested.

Comment 7: In the references, within article titles italicization (e.g. of *Caenorhabditis elegans*) is frequently missing. While this is often an artifact introduced by reference management software, it should be corrected in the final manuscript.

We thank the reviewer for all the helpful revision suggestions. We have made sure all the references are properly italicized where necessary.

**Reviewer 2:**
Comment 1: While FMO-4 is clearly placed in the ER calcium pathway genetically, the molecular mechanism by which FMO-4 would alter ER calcium is unclear. Notably, Tuckowski et al. highlight this gap in the discussion as well.

We thank the reviewer for identifying this important caveat. We hope to address the molecular mechanism by which FMO-4 alters ER calcium in upcoming projects.

Comment 2: Determining whether overexpression of catalytically dead FMO-4 or introduction of an inactivating point mutant into the endogenous locus phenocopy FMO-4 OE and KO animals would help distinguish between mechanisms involving protein-protein interactions or downstream metabolic regulation.

We thank the reviewer for this valuable suggestion. This is an experiment we are hoping to do in the near future to better understand molecular mechanisms and protein-protein interactions.

Reviewer 3:Comment 1: When measuring the effect of thapsigargin on development of fmo-4 mutants it would be great to use a developmental assay rather than quantifying normalized worm area. Also please add scale bars to Figure 3G and 4H, it seems that fmo-4 overexpression decreases worm size even in control conditions, clarify if this is the case.

We thank the reviewer for this feedback. In addition to quantifying normalized worm area in Figure 3G-I, we have added a developmental assay (Figure 3J) that shows the development time of wild-type worms on DMSO or thapsigargin as well as the *fmo-4* OE worms on DMSO or thapsigargin. These data validate that the *fmo-4* OE worm development is either delayed significantly or even prevented when the worms are treated with thapsigargin.

We have added scale bars to Figure 3G and 4H as suggested.

We also appreciate the reviewer’s observation of the *fmo-4* overexpression worms appearing smaller than wild-type worms in control conditions. We looked through the replicates and found that just one replicate showed a significant decrease in worm size, as observed in our unrevised manuscript. We repeated this experiment twice more to gather more data and determined that the *fmo-4* overexpression worms were ultimately not significantly different in size compared to wild-type worms. We have included the new images and quantifications in Figure 3G-I and Figure 4H-J in the revised manuscript.

Comment 2: correct or replace Supplementary Table 2, which is not showing a DAVID analysis as the title and text would suggest. We should see biological/molecular processes, effect sizes, p-values, ...

We thank the reviewer for identifying this issue. We have added more detail to the Supplementary Table 2 so that it is clearer what is being shown in each tab.

Comment 3: clarify the data presented in Supplementary Data 2 because it does not clearly explain what is shown

This is a great point, and we have added more detail to the Supplementary Data 2 to make sure the data are more clearly explained in each tab.

Comment 4: in Figure 5B the fluorescent images do not seem to reflect the quantification in panel 5C.

Thank you for this feedback. We re-analyzed our data to make sure the proper fluorescent images are included with their matching quantifications in Figure 5B-C.

Comment 5: where is Supplementary Data 3?

We thank the reviewer for noticing this. Supplementary Data 3 was accidentally missing from the first submission, and has now been added.

Comment 6: conceptually the last results section (regarding atf-6) does not add much to the story, I would consider removing these results

We appreciate this feedback. We have decided to keep Figure 7 because we think it helps to validate *fmo-4*’s role in calcium movement from the ER. While we show genetic interactions between *fmo-4* and key genes involved in calcium regulation (*crt-1, itr-1,* and *mcu-1*), we think that showing how *fmo-4* also interacts with *atf-6,* a known regulator of calcium homeostasis, strengthens and supports the genetic mechanisms of *fmo-4* proposed in this manuscript.

Comment 7: the model proposed in Figure 7E is not convincingly supported by the results:o the arrows connecting atf-6, fmo-4 and crt-1 (calreticulin) suggest that fmo-4 is downstream of atf-6 and upstream of crt-1: Berkowitz 2020 showed that atf-6 knockdown downregulates calreticulin, so unless the authors show that this downregulation is mediated directly by fmo-4, the more likely explanation is that atf-6 knockdown affects calcium levels which in turn induces fmo-4 expression.

We thank the reviewer for this helpful feedback. We have addressed this by updating our proposed model. We used a solid arrow leading from the reduction of *atf-6* to induction of *fmo-4,* as this is supported by our data in Figure 7A-B. We then used dashed arrows between *fmo-4* and *crt-1* as well as between *atf-6* and *crt-1* to indicate that more data is needed to clarify this part of the pathway.

Comment 8: Avoid pointing at a mitochondrial connection in the title as the only evidence supporting this interaction comes from the mcu-1 RNAi epistasis.

We appreciate the reviewer’s suggestion. We added another piece of evidence suggesting an interaction between *fmo-4* and the mitochondria to Supplementary Figure 7G-H. Here we show that while *fmo-4* OE worms are resistant to paraquat stress, knocking down *vdac-1* (a calcium regulator located in the outer mitochondrial membrane), abrogates this effect. We have kept mitochondria in our title but have made sure to temper our language in the main text to avoid pointing to a strong mitochondrial connection, since we have two pieces of evidence connecting *fmo-4* to the mitochondria.